# Multistage protective anti-CelTOS monoclonal antibodies with cross-species sterile protection against malaria

Wai Kwan Tang [1], Nichole D. Salinas[1], Surendra Kumar Kolli [2], Shulin Xu[2], Darya V. Urusova[3], Hirdesh Kumar[1], John R. Jimah[3,4], Pradeep Annamalai Subramani [2], Madison M. Ogbondah[2], Samantha J. Barnes[2], John H. Adams [2] & Niraj H. Tolia [1] ✉

CelTOS is a malaria vaccine antigen that is conserved in *Plasmodium* and other apicomplexan parasites and plays a role in cell-traversal. The structural basis and mechanisms of CelTOS-induced protective immunity to parasites are unknown. Here, CelTOS-specific monoclonal antibodies (mAbs) 7g7 and 4h12 demonstrated multistage activity, protecting against liver infection and preventing parasite transmission to mosquitoes. Both mAbs demonstrated cross-species activity with sterile protection against in vivo challenge with transgenic parasites containing either *P. falciparum* or *P. vivax* CelTOS, and with transmission reducing activity against *P. falciparum*. The mAbs prevented CelTOS-mediated pore formation providing insight into the protective mechanisms. X-ray crystallography and mutant-library epitope mapping revealed two distinct broadly conserved neutralizing epitopes. 7g7 bound to a parallel dimer of CelTOS, while 4h12 bound to a novel antiparallel dimer architecture. These findings inform the design of antibody therapies and vaccines and raise the prospect of a single intervention to simultaneously combat *P. falciparum* and *P. vivax* malaria.

Malaria caused by *Plasmodium* parasites continues to impose a major global health burden, afflicting 200 million people and causing half a million deaths annually[1]. Five *Plasmodium* species cause human malaria: *P. falciparum*, *P. vivax*, *P. knowlesi*, *P. ovale* and *P. malariae*[2]. These parasites coexist in endemic regions, and the highest health burden is attributed to *P. falciparum* and *P. vivax* infections[1,2]. Current malaria vaccines are only effective against a single *Plasmodium* species, and the lack of highly effective vaccines impedes the prevention and control of malaria[3–9]. Novel approaches to malaria vaccine development will complement ongoing efforts that have produced promising but less effective or short-lived protection against malaria[3,8–10]. Recent development of monoclonal antibodies to the

circumsporozoite protein (CSP) have shown proof of concept that antibody interventions and therapies can play a role in malaria control efforts[11,12]. Given reports of drug resistance for all existing classes of antimalarial medicines, new therapies and vaccines are needed for malaria control[13,14].

Cell-traversal protein for ookinetes and sporozoites (CelTOS) is a promising multistage transmission- and infection-blocking vaccine candidate for malaria. The malaria parasite life cycle can be broadly broken into three stages that are targets for intervention: initial infection of the liver in the human host[15,16], infection and reinvasion of red blood cells leading to the blood stage responsible for clinical manifestations of disease[17,18], and finally transmission to mosquitoes

[1]Host–Pathogen Interactions and Structural Vaccinology Section, Laboratory of Malaria Immunology and Vaccinology, National Institute of Allergy and Infectious Diseases, National Institutes of Health, Bethesda, MD, USA. [2]Center of Global Health and Interdisciplinary Research, College of Public Health, University of South Florida, Tampa, FL, USA. [3]Department of Molecular Microbiology, Washington University School of Medicine, St. Louis, MO, USA. [4]Present address: Department of Molecular Biology, Princeton University, Princeton, NJ, USA. ✉e-mail: niraj.tolia@nih.gov

leading to the next cycle of infection[19]. CelTOS is a unique multistage *Plasmodium* antigen that plays a critical cell-traversal role during infection of the human liver and during parasite transmission to the mosquito vector[20]. CelTOS is secreted by *Plasmodium* parasites and is localized to the surface of sporozoites[20]. Despite high sequence conservation within *Plasmodium* and apicomplexan parasites, CelTOS does not share significant similarity to other proteins of known function[21]. Roles for CelTOS in pore formation during cell traversal[21–24] and in gliding motility of the sporozoite have been reported[25]. The crystal structure of *P. vivax* CelTOS revealed resemblance to membrane-disrupting proteins and led to the demonstration that CelTOS is a phosphatidic acid-specific pore-forming protein[21–24]. CelTOS is proposed to disrupt the plasma membrane of invaded cells, specifically those of the mosquito midgut epithelium and mammalian liver sinusoidal layer, and enables the exit of parasites during cell traversal[21]. Pore formation is dependent on lipid binding, dimer dissociation, and dramatic conformation changes in CelTOS[24].

CelTOS is an antigen that is preferentially recognized by the immune response of human volunteers vaccinated with irradiated sporozoites[26]. Humans naturally develop antibodies to CelTOS after infection, although antibody levels are low[27]. Immunization with *P. falciparum* CelTOS partially protects against heterologous challenge with *P. berghei*, suggesting a role for CelTOS as an anti-infective vaccine for malaria[28]. Polyclonal antibodies to CelTOS inhibit oocyst development of *P. falciparum* gametocytes and reduce transmission[29]. The passive transfer of polyclonal antibodies from CelTOS-immunized rabbits protects naïve mice from malaria infection, demonstrating the infection-blocking capability of CelTOS antibodies[30]. Both studies[29,30] reported that CelTOS polyclonal antibodies showed incomplete protection against malaria transmission and infection. Immunization with *P. vivax* CelTOS[31] did not elicit protection upon challenge with transgenic *P. berghei* parasites expressing *P. falciparum* or *P. vivax* CelTOS, suggesting that CelTOS elicits a predominantly nonprotective response. Virus-like particle display of the C-terminal domains of *P. falciparum* CelTOS was needed to produce functional immunity against *P. berghei* sporozoite infection[32]. Antigens have the propensity to elicit both neutralizing human antibodies that protect against disease and interfering human antibodies that interfere with the function of neutralizing antibodies through an immune evasion mechanism called antigenic diversion[33]. CelTOS immunogens that predominantly elicit a protective antibody response will likely improve protective outcomes, and it is necessary to identify epitopes in CelTOS that are targeted by neutralizing and/or protective antibodies to inform the design of effective CelTOS immunogens and monoclonal antibody interventions.

In this study, we report distinct monoclonal antibodies raised against *P. falciparum* and *P. vivax* CelTOS that confer cross-species sterile protection to mice and block transmission of *P. falciparum* parasites by preventing oocyst formation. These findings will inform the design of immunogens to be used in cross-species CelTOS-based infection- and transmission-blocking malaria vaccines. Monoclonal antibodies have recently demonstrated potential as prophylactic treatments for malaria[11,12,28,29,34–36], and this work forms a strong foundation for the further development of cross-species protective and multistage monoclonal antibodies that target CelTOS.

## Results
### Development of monoclonal antibodies
A panel of monoclonal antibodies (mAbs) was isolated from mice immunized with either *P. vivax* CelTOS (PvCelTOS) or *P. falciparum* CelTOS (PfCelTOS). Six mAbs (7g7, 8c2, 6c4, 8d2, 6e11 and 6f11) were isolated from immunization with PvCelTOS, and three mAbs (4h12, 4d1 and 2f11) were isolated from immunization with PfCelTOS. These mAbs were initially identified by ELISA screening of hybridoma supernatants followed by hybridoma subcloning to produce the final mAbs that were evaluated for binding to CelTOS proteins using a dose dependent

ELISA with purified IgG (Fig. 1a, b). Five of the six ELISA positive mAbs (7g7, 8c2, 6c4, 8d2 and 6e11) isolated from mice immunized with PvCelTOS interacted strongly with PvCelTOS (Fig. 1a). Subsequent scale-up of hybridoma culture of 6c4 showed that the cell line was unstable in expressing IgG. Therefore, the weak binder 6f11 and unstable 6c4 were omitted from further study. Epitope binning of the four anti-PvCelTOS mAbs (7g7, 8c2, 8d2 and 6e11) showed that 7g7 and 8d2 shared epitopes that did not compete with the epitopes of 8c2 and 6e11 (Fig. 1c). Two out of three ELISA positive mAbs (4h12 and 4d1) isolated from mice immunized with PfCelTOS interacted strongly with the PfCelTOS protein (Fig. 1b) and were prioritized for further analysis. These two anti-PfCelTOS mAbs shared a competing epitope (Fig. 1d). All anti-CelTOS mAbs were confirmed to belong to the IgG1 subclass by an isotyping ELISA.

### Loop 5 in PvCelTOS is the key epitope for 7g7
We determined the binding epitope of 7g7 on PvCelTOS by solving the crystal structure of the antibody-antigen complex. A 3.2 Å resolution structure of the 7g7 Fab fragment in complex with PvCelTOS (Supplementary Table 1) revealed a 2:2 complex of Fab and PvCelTOS with two monomers of PvCelTOS arranged in parallel to form a dimer. Each monomer of PvCelTOS interacted with one molecule of the 7g7 Fab (Fig. 2a). The two PvCelTOS monomers in the antibody-bound structure were highly similar, with a root mean square deviation (RMSD) of 0.27 Å across the Cα atoms of 122 residues.

PvCelTOS is observed as a dimer in the 7g7 antigen-antibody complex. The dimeric architecture is consistent with the dimer observed for unbound PvCelTOS[21]. Binding of 7g7 did not induce major structural changes in PvCelTOS, as superposition of the PvCelTOS monomer and dimer from the current study with the published PvCelTOS apo structure (PDB:5TSZ)[21] revealed RMSD values of 0.75 Å and 0.88 Å, respectively, across all Cα atoms (Fig. 2b, c).

The epitope for 7g7 in PvCelTOS has a buried surface area of 694 Å between the antibody and the antigen. The interactions between 7g7 and PvCelTOS arose predominantly from the antibody heavy chain, with all three heavy chain CDRs contacting PvCelTOS (Supplementary Fig. 1a, Supplementary Table 2). In contrast, very limited interactions were observed from the antibody light chain. The conformational epitope in PvCelTOS consists of a helix-loop-helix structure comprised of helix 2 (H2), helix 3 (H3), and loop 5. All the hydrogen bonds and salt bridges between 7g7 and PvCelTOS arise from Loop 5 (Fig. 2d, e, Supplementary Fig. 1a, Supplementary Table 2). The binding epitope on PvCelTOS was confirmed by mutant library epitope mapping, where alanine mutations were introduced into 3-4 short segment residues along the surface of PvCelTOS. All four mutants carrying mutations within Loop 5 failed to interact with 7g7 (Fig. 2f). These results established loop 5 of PvCelTOS as the key binding epitope of 7g7. The binding affinity of 7g7 to PvCelTOS was determined by biolayer interferometry (BLI) with the mAb immobilized on the biosensor. 7g7 bound PvCelTOS with a $K_D$ value of 50.73 nM (Fig. 2g, Supplementary Table 4).

### Antibody 4h12 binds to Loop 3 in PfCelTOS
A second mouse monoclonal antibody, 4h12, was obtained by immunization with PfCelTOS. We solved the crystal structure of PfCelTOS in complex with the 4h12 Fab to 3.5 Å resolution (Supplementary Table 1). Molecular replacement using a 4h12 Fab model readily resulted in a structural solution with well-defined electron density around the entire 4h12 Fab. Additional density that resembled two helixes of PfCelTOS was observed that contacted the Fabs. This structure revealed that two molecules of PfCelTOS comprised residues S60-S116 bound to two molecules of 4h12 (Fig. 3a). Each molecule of PfCelTOS bound to one 4h12 Fab, resulting in a buried surface area of 612 Å². The antigen-antibody complex is stabilized by hydrophobic interactions and hydrogen bonding between the heavy chain of 4h12 and Loop 3

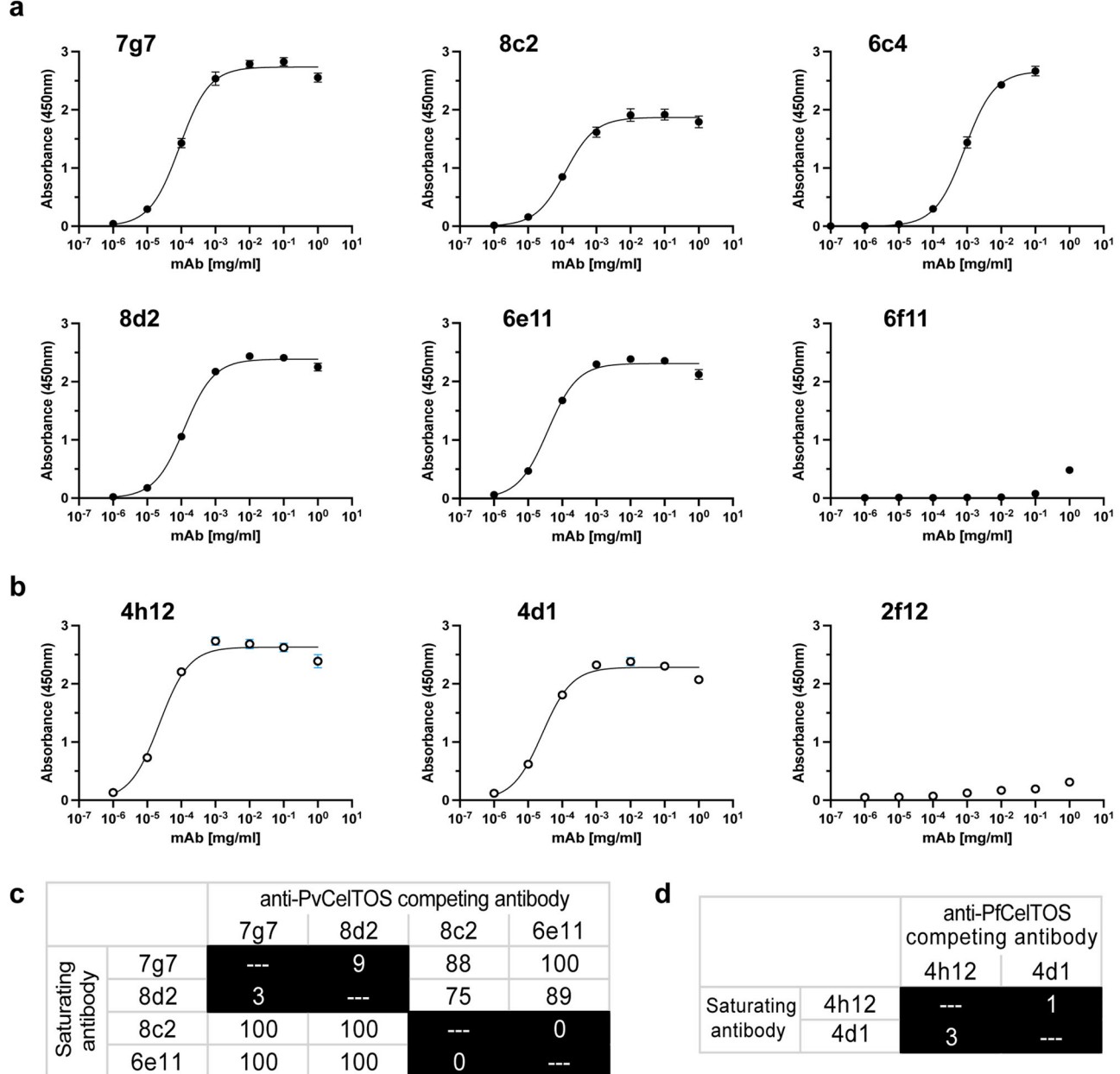

**Fig. 1 | Monoclonal antibodies isolated from mice immunized with CelTOS.** Binding of purified IgGs to **a** PvCelTOS and **b** PfCelTOS. The antigen-antibody interactions were investigated using ELISA. Serial tenfold dilutions of the IgGs were allowed to interact with CelTOS proteins coated on the plates. The mean ± s.d. of five replicates are shown. Epitope binning of **c** anti-PvCelTOS IgGs and **d** anti-PfCelTOS IgGs. Biotinylated CelTOS was immobilized on biosensors. Saturating IgGs are listed on the left and the competing IgGs are listed on the top. Reported scores are a percentage of total binding response of that IgG in the absence of competing IgG. Values < 50% were shaded in black. Any experiments with >100% and <0% were given a score of 100 and 0, respectively.

(residues K82-E89) of PfCelTOS (Fig. 3b, Supplementary Fig. 1b, Supplementary Table 3).

The binding of PfCelTOS to 4h12 showed a $K_D$ value of 65.4 nM, which was similar to the affinity observed for PvCelTOS to 7g7 (Fig. 3c, Supplementary Table 4). PfCelTOS in the 4h12 antibody-bound structure appeared highly flexible, and only 57 residues (S60-S116) covering the H2-loop3-H3 (helix2-loop3-helix3) region were ordered (Fig. 3b, 3d). Mutant library epitope mapping independently confirmed this segment of PfCelTOS as critical for binding to 4h12. There was a clear loss of interaction between 4h12 and mutants carrying alanine amino acid changes within the Loop 3 segment of PfCelTOS (Fig. 3e). Together, these results demonstrated that 4h12 bound an epitope distinct from the epitope recognized by 7g7.

## 4h12 traps CelTOS in a unique oligomeric state with a reordered dimer interface

CelTOS forms a parallel dimer in both the unbound PvCelTOS structure (PDB:5TSZ)[21] and the 7g7 bound structure (Fig. 2a). The binding of 7g7 to loop 5, which is located on the side of the PvCelTOS dimer, allows two 7g7 antibodies to simultaneously interact with the PvCelTOS dimer. This parallel dimer architecture placed loop 3 of each monomer in proximity, pointing in the same direction (Fig. 2c). In contrast, the PfCelTOS-4h12 structure revealed that the PfCelTOS dimer in the bound structure was organized in an antiparallel manner, placing the binding epitope of 4h12 at loop 3 of PfCelTOS at the two opposing ends of the dimer and allowing a 2:2 binding ratio of the PfCelTOS monomer to 4h12 (Fig. 3a, b).

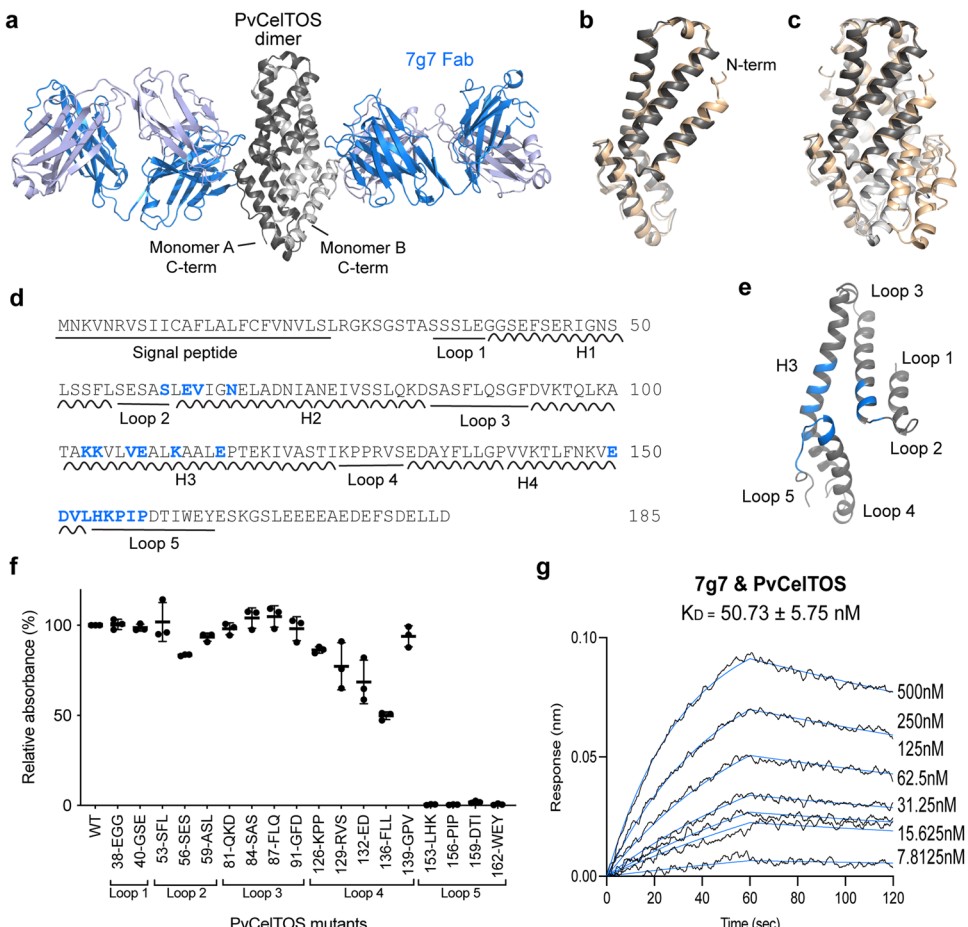

**Fig. 2 | Binding of 7g7 on PvCelTOS. a** Crystal structure of PvCelTOS dimer in complex with 7g7 Fab. The dimer of PvCelTOS is in black/gray ribbons; the heavy and light chains of 7g7 Fab are in blue and light blue ribbons. Superposition of PvCelTOS **b** monomer and **c** dimer from the current study (black/gray) with the previously reported PDB:5TSZ[21] (wheat). **d** Full-length sequence of PvCelTOS. Helix and loop regions are labeled. Binding interface residues are in blue. **e** Binding epitope of 7g7 on PvCelTOS. The PvCelTOS monomer is represented in gray ribbon; the binding interface is highlighted in blue. **f** Mapping of epitope within the loop regions using ELISA. Each PvCelTOS mutant carries consecutive 3-4 residues mutated to alanine and the proteins were purified for ELISA to examine their interaction with 7g7. The graph is shown as mean ± s.d. of three biological replicates. **g** Binding affinity of 7g7 to PvCelTOS as determined by BLI. Representative sensorgram showing association and dissociation when 7g7 is immobilized on biosensors. Black lines are sensograms for two-fold dilution series of PvCelTOS. Blue lines are fitted curves correspond to a 1:1 model. The $K_D$ value is the mean ± s.e.m. of three biological replicates.

The PvCelTOS and PfCelTOS proteins share a high sequence identity of 44.5% (Fig. 3h). Although the structure of PfCelTOS alone has not been reported, it is expected to be similar to PvCelTOS based on sequence similarity and consistent with Alphafold predictions (https://alphafold.ebi.ac.uk/entry/A0A2R4QLI0). Strikingly, the same residues that comprised the dimer interface in PvCelTOS were used to create the alternate dimer interface in the PfCelTOS-4h12 complex, although with a different interaction matrix (Fig. 3f–h). This suggested that the dimer interface of CelTOS is plastic and can accommodate both parallel and antiparallel architectures. The conformational flexibility is consistent with the observations that rigidification of CelTOS prevents function and that conformational flexibility is necessary for activity[24].

## 7g7 and 4h12 inhibit CelTOS pore formation

CelTOS disrupts phosphatidic acid-containing membranes[21–24]. We investigated the ability of 7g7 and 4h12 to inhibit PvCelTOS- and PfCelTOS-mediated disruption of liposomes containing phosphatidic acid. The assays were established with liposomes in excess of CelTOS, resulting in maximal CelTOS-dependent liposome disruption of between 30 and 70%. This is to mimic the physiological conditions where the phosphatidic acid lipid membranes are in far excess of the available CelTOS molecules on a sporozoite. In the presence of the CelTOS protein alone, a maximum of ~35% and ~50% liposome disruption was observed for PvCelTOS and PfCelTOS, respectively (Fig. 4a, b). A concentration-dependent inhibition of liposome disruption by CelTOS was observed upon increasing concentration of 7g7 or 4h12 antibodies (Fig. 4). Significant inhibition was observed at the higher concentrations tested (Supplementary Fig. 2). These data suggest that binding of 7g7 to PvCelTOS and 4h12 to PfCelTOS can prevent pore formation and provides one possible mechanism of action of these mAbs in infection and transmission blocking.

## mAbs cross-react with both PvCelTOS and PfCelTOS

We examined whether the two mAbs cross-react with CelTOS protein from different *Plasmodium* species given the high sequence identity and similarity between PvCelTOS and PfCelTOS (Fig. 5). While the binding epitope of 4h12 to PfCelTOS is mostly formed by the backbone hydrophobic interaction and hydrogen bond with the heavy chain of 4h12, the side chains of K82 and F87 play a key role in forming salt bridges and pi stacking with the heavy chain residues of 4h12, respectively. This binding epitope is highly conserved among *P. falciparum* and *P. vivax* (Fig. 3h), which explains the cross-reactivity of 4h12 among the two species of *Plasmodium*. As determined by BLI, the cross-reactive binding affinity of 4h12 with PvCelTOS was 51.81 µM, which was ~1000-fold higher $K_D$ than that observed with PfCelTOS

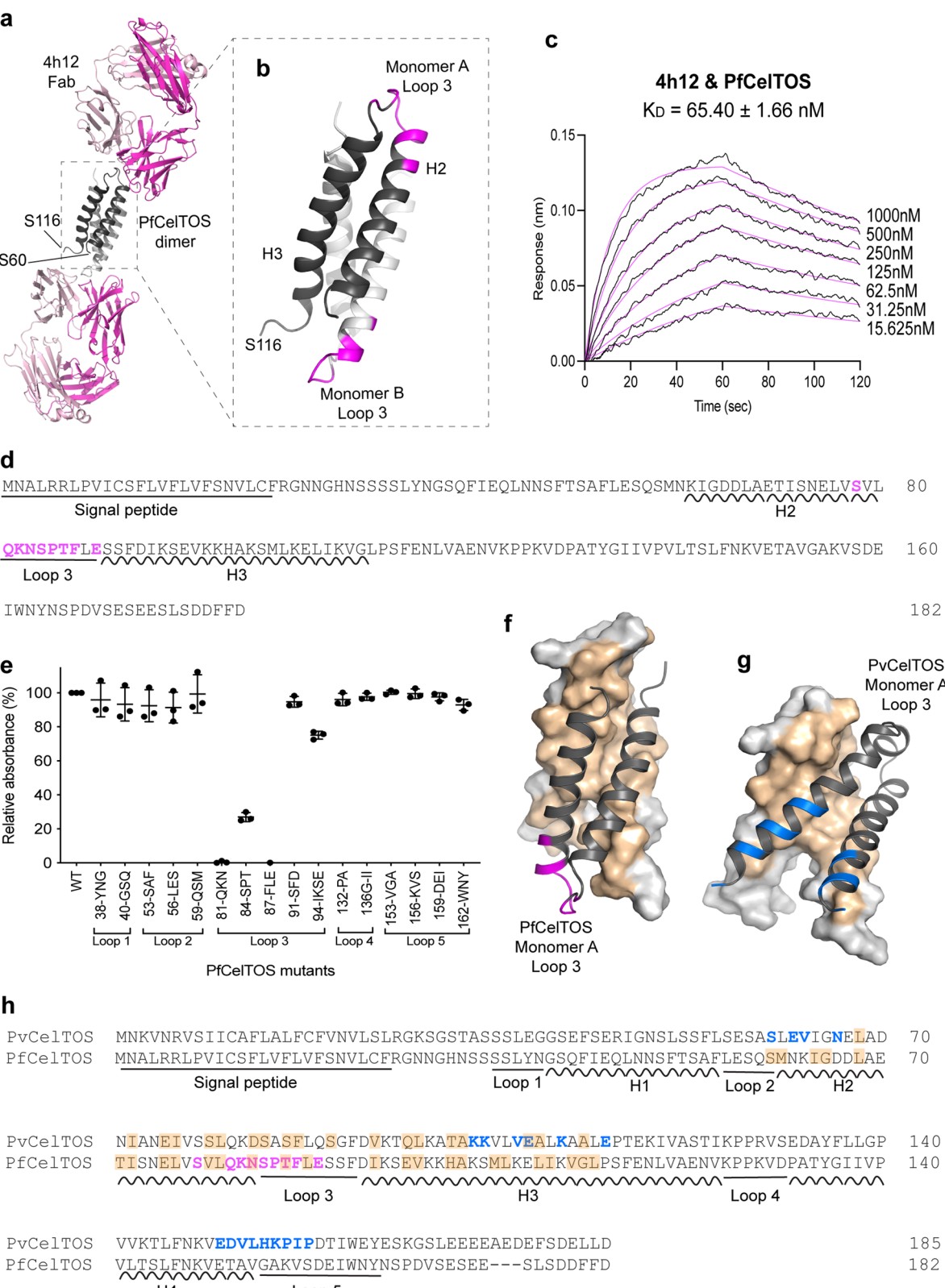

**d**

MNALRRLPVICSFLVFLVFSNVLCFRGNNGHNSSSSLYNGSQFIEQLNNSFTSAFLESQSMNKIGDDLAETISNELVSVL        80
<u>————————————————————————</u>                                                    H2
Signal peptide

QKNSPTFLESSFDIKSEVKKHAKSMLKELIKVGLPSFENLVAENVKPPKVDPATYGIIVPVLTSLFNKVETAVGAKVSDE        160
Loop 3              H3

IWNYNSPDVSESEESLSDDFFD                                                                182

**h**

```
PvCelTOS  MNKVNRVSIICAFLALFCFVNVLSLRGKSGSTASSSLEGGSEFSERIGNSLSSFLSESASLEVIGNELAD        70
PfCelTOS  MNALRRLPVICSFLVFLVFSNVLCFRGNNGHNSSSSLYNGSQFIEQLNNSFTSAFLESQSMNKIGDDLAE        70
          ———————————————————————               Loop 1    H1    Loop 2   H2
               Signal peptide

PvCelTOS  NIANEIVSSLQKDSASFLQSGFDVKTQLKATAKKVLVEALKAALEPTEKIVASTIKPPRVSEDAYFLLGP        140
PfCelTOS  TISNELVSVLQKNSPTFLESSFDIKSEVKKHAKSMLKELIKVGLPSFENLVAENVKPPKVDPATYGIIVP        140
                       Loop 3           H3              Loop 4

PvCelTOS  VVKTLFNKVEDVLHKPIPDTIWEYESKGSLEEEEAEDEFSDELLD                            185
PfCelTOS  VLTSLFNKVETAVGAKVSDEIWNYNSPDVSESEE---SLSDDFFD                            182
          H4              Loop 5
```

(65.40 nM) (Fig. 5a and Supplementary Table 4). Similarly, 7g7 bound to PvCelTOS with a $K_D$ value of 50.7 nM and cross-reacted with PfCelTOS with an -10-fold higher $K_D$ value of 0.48 μM (Fig. 5b and Supplementary Table 4).

Our structural data demonstrated CelTOS is conformational flexible. The architecture of CelTOS is believed to be vary between the pre-pore state arranged as an array on the surface of the parasite and pore state

upon membrane insertion[24]. As the binding affinities reported from BLI can only reflect the interaction of antibodies with the soluble CelTOS dimer and cannot account for avidity affects caused by repetitive expression on the parasite surface, we investigated the binding and cross reactivity of the antibodies to sporozoites. Cryopreserved *P. falciparum* and *P. vivax* sporozoites were stained effectively well by both 7g7 and 4h12 demonstrating these antibodies bind and recognize CelTOS from

**Fig. 3 | Binding of 4h12 on PfCelTOS. a** Crystal structure of PfCelTOS in complex with 4h12 Fab. The PfCelTOS dimer is in black/gray ribbons; the heavy and light chains of 4h12 Fab are in magenta and pink ribbons. **b** Binding epitope of 4h12 on PfCelTOS. The PfCelTOS dimer is represented in black/gray ribbon, the binding interface is highlighted in magenta. **c** Binding affinity of 4h12 to PfCelTOS as determined by BLI. Representative sensorgram showing association and dissociation with 4h12 immobilized on biosensors. Black lines are sensorgrams for two-fold dilution series of PfCelTOS. Magenta lines are fitted curves correspond to a 1:1 model. The $K_D$ value is the mean ± s.e.m. from three biological replicates. **d** Full-length sequence of PfCelTOS. Helix and loop regions modeled in the structure were labeled. The binding interface residues are in magenta. **e** Mapping of epitope within the loop regions using ELISA. Each PfCelTOS mutant carries consecutive 3-4 residues mutated to alanine and the proteins were purified for ELISA to examine their

interaction with 4h12. The graph is shown as mean ± s.d. of three biological replicates. **f, g** Comparison of the dimer orientations between PfCelTOS and PvCelTOS. Monomer A is represented in black ribbons. Monomer B is represented in gray surfaces, the dimer interface on monomer B is colored in wheat. The monomer B in both dimer structures is shown in the same orientation by superposition.
**f** The dimer of PfCelTOS with the 4h12 binding epitope highlighted in magenta.
**g** Only residues S71-E126 of the dimer of PvCelTOS which correspond to the resolved segment in the PfCelTOS structure are shown. The 7g7 binding epitope is highlighted in blue. **h** Sequence alignment of PvCelTOS and PfCelTOS. Helix and loop regions based on the structure of PvCelTOS are labeled. The binding interface residues of 7g7 and 4h12 are in blue and magenta, respectively; the dimer interface residues are highlighted in wheat.

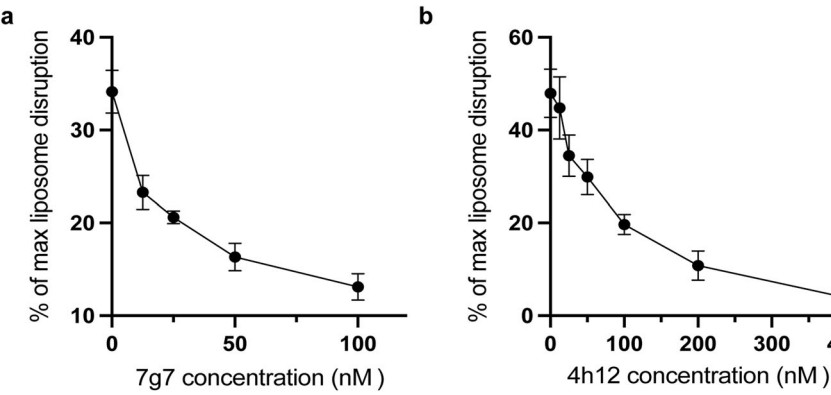

**Fig. 4 | 7g7 and 4h12 inhibited CelTOS-mediated membrane disruption.** The binding of **a** 7g7 to PvCelTOS and **b** 4h12 to PfCelTOS showed a concentration dependent inhibition of liposome disruption by CelTOS. The mean ± s.e.m from five replicates are shown.

different species (Fig. 5c, d). Similarly, we also observed cross reactions of the antibodies to the transgenic *P. berghei* sporozoites expressing either PfCelTOS or PvCelTOS (Fig. 5e, f). Together, the data demonstrate the cross-species reactivity of 7g7 and 4h12 to *P. falciparum* and *P. vivax* parasites.

### 7g7 and 4h12 block transmission of *P. falciparum* parasites

The ability of monoclonal antibodies to block the transmission of sexual blood stage *P. falciparum* to mosquito stage oocysts was investigated by a standard membrane feeding assay (SMFA). This is an established method to evaluate the transmission-reducing potential of antibodies. Transgenic *P. falciparum* parasites expressing luciferase were developed to evaluate the transmission-reducing activity of the mAbs. In the presence of 400 μg/ml mAb in the blood meal, significantly lower mosquito RLU values were obtained for 7g7 (538.5 ± 468.8) and 4h12 (488.2 ± 507.8) when compared to the mouse IgG1 control group (1550 ± 1408)[37] and no antibody control group (1336 ± 1005) (Fig. 6). These results demonstrated that both 7g7 and 4h12 could reduce the transmission of *P. falciparum* parasites to mosquitoes. In addition, the data were consistent with the cross-species activity of 7g7 against *P. falciparum*.

### Cross-species sterile protection against challenge with transgenic parasites containing *P. falciparum* or *P. vivax* CelTOS

CelTOS is required for parasite traversal through host cells, and traversal is required for establishment in the liver and transmission to the mosquito vector. The functional protection against liver stage establishment by 7g7 and 4h12 in vivo was assessed in mice using transgenic *P. berghei* sporozoites in which the coding sequence of PbCelTOS was replaced by either PvCelTOS (Pb-PvCelTOS) or PfCelTOS (Pb-PfCelTOS). Mice were first passively immunized with 580 μg of either 7g7 or 4h12 antibody intraperitoneally and challenged by intravenous injection of 200 transgenic sporozoites expressing either PvCelTOS or

PfCelTOS. Liver stage infectivity was evaluated by in vivo imaging of luciferase expression in the liver (Supplementary Fig. 3). One hundred percent of animals in the no antibody and isotype matched control groups were infected 44 hrs post challenge (Fig. 7). The parasite liver burden recorded as relative luciferase units (RLU) revealed that mice immunized with 7g7 or 4h12 had significantly lower parasite liver burdens than control mice that received no antibody (Fig. 7a, c). Strikingly, significantly lower liver burdens were also observed by both mAbs with either Pb-PfCelTOS or Pb-PvCelTOS transgenic parasites, demonstrating cross-species protection in vivo.

We further evaluated blood-stage parasitemia to examine sterile protection induced by passive immunization with 7g7 and 4h12 (Fig. 7b, d). One hundred percent of the animals in the no antibody and isotype matched control groups[38] demonstrated blood stage parasitemia by day 7 (Fig. 7). In contrast, 4h12 and 7g7 showed 100% sterile protection when challenged with Pb-PfCelTOS sporozoites (Fig. 7b). Similarly, 4h12 showed 100% sterile protection and 7g7 demonstrated 90% sterile protection with normal patency of parasitemia compared to the control when challenged with Pb-PvCelTOS sporozoites (Fig. 7d). These data suggest that both 7g7 and 4h12 could cross-react with both PvCelTOS and PfCelTOS and provide cross-species sterile protection against malaria.

### Discussion

CelTOS is a unique malaria vaccine candidate because it is conserved across *Plasmodium* and other apicomplexan parasites, including those that cause human malaria[21], and because it plays a key role in both the liver stage and mosquito stage of the malaria parasite life cycle[20]. CelTOS is a target of vaccine-induced and naturally acquired immunity to sporozoites in humans[26,27]. Additionally, PvCelTOS from field isolates exhibits a low level of polymorphisms that do not coincide within the key binding epitope of 7g7[39,40] (Supplementary Fig. 4). Similarly, the binding epitope of 4h12 on PfCelTOS did not overlap with the 34

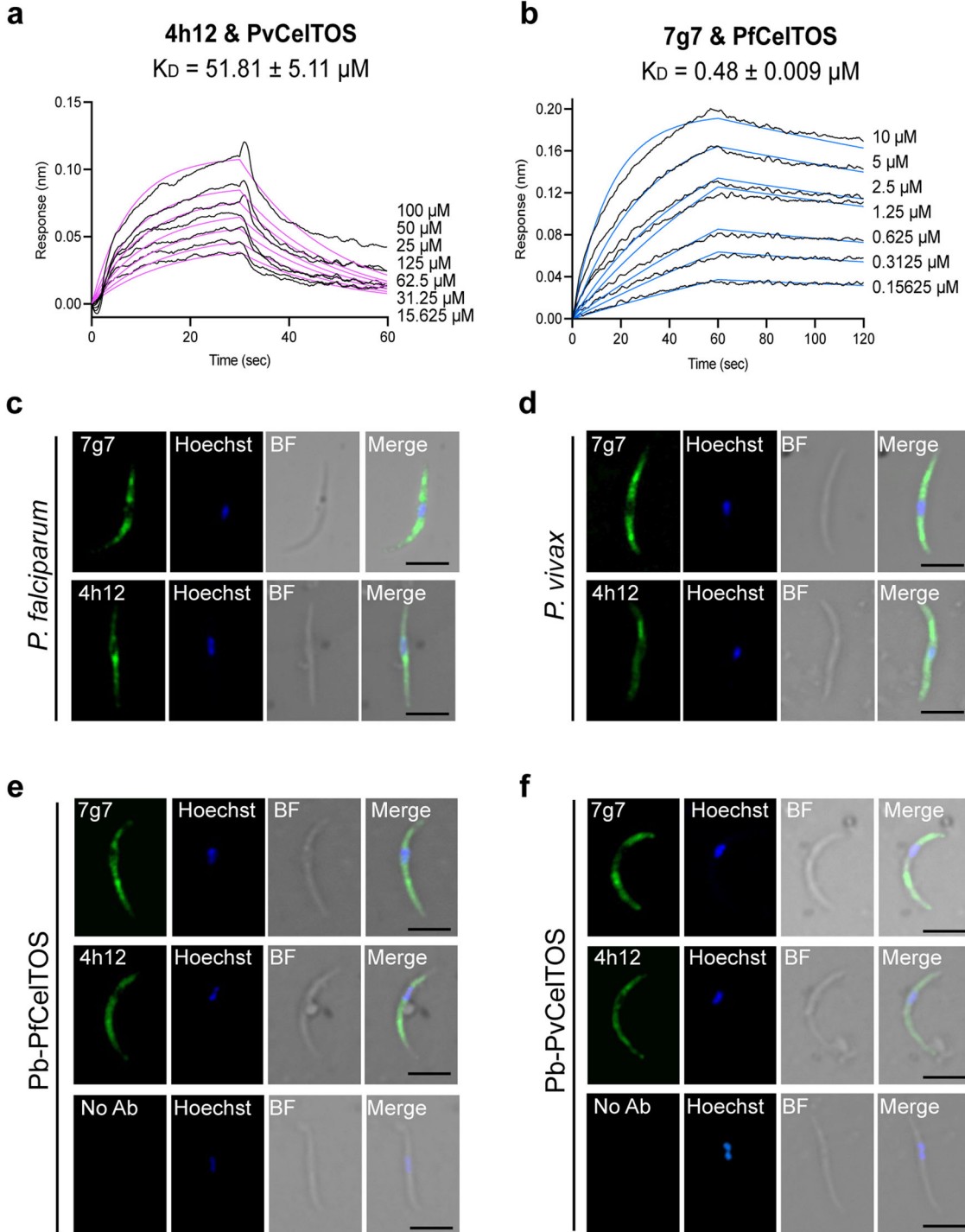

**Fig. 5 | Cross reactive binding of mAbs to CelTOS.** Binding affinities of **a** 4h12 to PvCelTOS and **b** 7g7 to PvCelTOS as determined by BLI. Representative sensorgrams showing association and dissociation with antibodies immobilized on biosensors. Black lines are sensorgrams for two-fold dilution series of CelTOS. Blue and magenta lines are fitted curves corresponding to a 1:1 model. The $K_D$ values are the mean ± s.e.m. from three biological replicates. **c, d** Indirect immunofluorescence of *Plasmodium* sporozoites with antibodies. Cryopreserved *P. falciparum* and *P. vivax* sporozoites were stained with monoclonal antibodies 7g7 and 4h12. One representative image from two independent replicates with similar results is shown. **e, f** Transgenic *P. berghei* sporozoites expressing *P. falciparum* CelTOS (Pb-PfCelTOS) or *P. vivax* CelTOS (Pb-PvCelTOS) were stained with monoclonal antibodies 7g7 and 4h12. Immunoreactivity was revealed with donkey anti-mouse Alexa flour Plus 488 (green). Nuclei are stained with Hoechst 33342 (blue). BF, bright field; No Ab, no primary antibody control (Scale bars, 5 μm). One representative image from three independent replicates with similar results is shown.

known nonsynonymous SNPs reported in the MalariaGEN Pf4 dataset (Supplementary Fig. 5). Immunization with CelTOS is therefore likely to provide protection from multiple circulating parasite strains. CelTOS has been investigated in preclinical studies as a potential vaccine candidate for malaria with varying levels of success[28–32]. To improve

the quality of the protective response through immunogen design, it is necessary to generate antibodies against CelTOS and identify their biological functions and binding epitopes. This study reported two monoclonal antibodies raised against *P. falciparum* and *P. vivax* CelTOS that revealed distinct binding epitopes, inhibited pore formation,

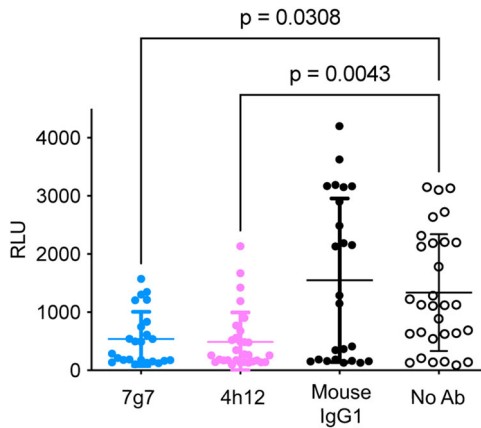

**Fig. 6 | Both 7g7 and 4h12 inhibited *P. falciparum* oocyst development in SMFA.** Mosquito luciferase assay demonstrated 7g7 and 4h12 were able to inhibit oocyst development in mosquito. Antibodies at 400 µg/ml were included in the mosquito blood meal. Mouse IgG1 3A4, an anti-PvDBP (*P. vivax* Duffy binding protein antibody)[37], was used as a negative antibody control and no antibody (No Ab) as a negative control. Inhibition ability was measured by luciferase activity (RLU values) of infected mosquitoes. The graph is shown as mean ± s.d. of RLU values from 24, 30, 30 and 24 replicates for groups 7g7, 4h12, no antibody and anti-PvDBP, respectively. Statistical differences were calculated using Kruskal–Wallis and Dunn's multiple comparison test.

reduced *P. falciparum* parasite transmission, and conferred cross-species sterile protection in mice.

Crystal structures of the mAbs in complex with CelTOS revealed the conformational neutralizing epitopes. These epitopes were independently identified by mutant library epitope mapping. Two distinct nonoverlapping epitopes were identified that can independently elicit transmission blocking activity and cross-species sterile protection. The disparate locations of the 7g7 and 4h12 epitopes in CelTOS suggested that the full-length CelTOS protein should be prioritized for further vaccine development and that focusing the immune response to one epitope would eliminate the benefits of retaining the second epitope.

Interestingly, the multiple available structures of CelTOS demonstrate plasticity in the oligomeric architecture of the protein. CelTOS dimer dissociation and conformational flexibility are required for CelTOS activity[24]. The dimeric architecture observed in both PfCelTOS and PvCelTOS reported here are conformational states prior to membrane insertion and pore formation and are therefore termed prepore states. The data presented here suggested that 7g7 and 4h12 bound to the prepore states of CelTOS and trapped CelTOS in an inactive form on the surface of sporozoites. This approach is analogous to antibody-mediated targeting of the prefusion states of viral proteins that have met the high vaccine efficacy of multiple viral vaccines[41–44]. Together, targeting the prepore states of CelTOS appears to be a feasible mechanism for antibody-dependent protection against malaria and apicomplexan parasites.

Biophysical studies have suggested that these mAbs protect against malaria parasites by impairing CelTOS-mediated membrane disruption. However, other mechanisms are also likely to result in the observed sterile protection and transmission blocking activity. For instance, antibody-mediated effector functions may contribute to the clearance of sporozoites in vivo[45–48]. Binding of 7g7 and 4h12 to the prepore state of CelTOS on the surface of sporozoites in an extracellular environment may facilitate antibody-dependent phagocytosis or antibody-dependent cellular cytotoxicity in vivo. Similarly, complement deposition plays an important role during the elimination of parasites in the mosquito midgut, and mAbs may facilitate complement activation and deposition to prevent transmission and/or infection.

The most advanced liver-stage infection blocking vaccines for malaria, RTS,S and R21, are based on the circumsporozoite protein of *P. falciparum*[5–9]. These vaccines have shown moderate protective efficacy and demonstrate that malaria vaccines can reduce disease burdens in endemic areas[7]. Similarly, monoclonal antibodies that target CSP have shown high efficacy as prophylactic antibody treatments to prevent malaria infection in endemic areas[11,12]. However, these vaccines and antibodies are specific to *P. falciparum* malaria and do not protect against other *Plasmodium* parasites that cause human malaria. In addition, these vaccines and antibodies can prevent infection of the liver but have no effect on other stages of the parasite life cycle. The data presented demonstrate that both 7g7 and 4h12 antibodies can effectively bind to *P. falciparum* and *P. vivax* CelTOS and recognize *P. falciparum* and *P. vivax* sporozoites in a cross-species manner. These antibodies provided cross-species sterile protection in challenge study and cross-species transmission reducing activity. Our data suggested that future CelTOS-based malaria vaccines and antibodies have major advantages in cross-species protection and may elicit both infection- and transmission-blocking activity. It is also plausible that a combination of CSP-based and CelTOS-based vaccines and antibodies may lead to additive or synergistic protection to combat malaria.

Structural vaccinology holds promise as a method to improve malaria vaccine design[49]. This process relies on identifying neutralizing and non-neutralizing epitope maps in antigens[15,33,50–58] and applying this information to design immunogens that elicit protective antibody responses[49,59–65]. It is also necessary to identify and pursue new antigens that may lead to potent and durable multistage cross-species protection. This study provides proof-of-concept that a single antibody against CelTOS can block multiple stages of the parasite life cycle, can simultaneously confer cross-species protection against both *P. falciparum* and *P. vivax*, and identifies the linear and three-dimensional structures of neutralizing epitopes in CelTOS. These findings will inform the design of immunogens to be used in cross-species CelTOS-based infection- and transmission-blocking malaria vaccines and form a strong foundation for the further development of cross-species protective and multistage monoclonal antibodies that simultaneously protect against *P. falciparum* and *P. vivax*.

## Methods
### Expression and purification of CelTOS
PfCelTOS (3D7, PF3D7_1216600, residue F25-D182) and PvCelTOS (Sal-1, PVX_123510, residue L36-D196) constructs were expressed and purified as previously described[21]. Briefly, CelTOS cloned into pET28 vector was transformed into BL21(DE3) cells and grown at 37 °C until $A_{600}$ reached ~0.6. Expression was induced with 1 mM IPTG and continued to grow at 37 °C for 4 h. Cell pellets harvested were resuspended either in 25 mM Tris, pH 7.4, 0.3 M NaCl supplemented with 30 mM imidazole for purification by Ni Sepharose excel (Cytiva, Marlborough, MA) or 50 mM Tris pH 8, 100 mM NaCl, 5 mM DTT, 0.25 mg/mL Lysozyme (Sigma Aldrich, St. Louis, MO) for purification by Nickel HTC Agarose resin (Goldbio, St. Louis, MO). Cells were lysed by sonication and centrifuged at 39,000 x *g* for 20 min. The lysate was then loaded onto either Ni Sepharose excel or Nickel HTC Agarose resin for purification. For the Ni Sepharose excel, the resin was washed with 25 mM Tris, pH 7.4, 0.3 M NaCl supplemented with 30 mM imidazole and CelTOS protein was eluted with 25 mM Tris, pH 7.4, 0.3 M NaCl supplemented with 150 mM imidazole. For the Nickel HTC Agarose resin, the resin was washed with 50 mM Tris, pH 8, 0.1 M NaCl supplemented with 10 mM imidazole and CelTOS protein was eluted with 30 mM Tris, pH 8, 0.1 M NaCl supplemented with 500 mM imidazole. Fractions from the Nickel resin purification containing CelTOS were concentrated and loaded into Superdex 200 increase column (Cytiva, Marlborough, MA) equilibrated with 20 mM Tris, pH 8.0, 100 mM NaCl for crystallography, PBS for animal studies, or 10 mM HEPES pH 7.4 150 mM NaCl 3 mM EDTA for BLI. Fractions were pooled and concentrated and stored at −80 °C.

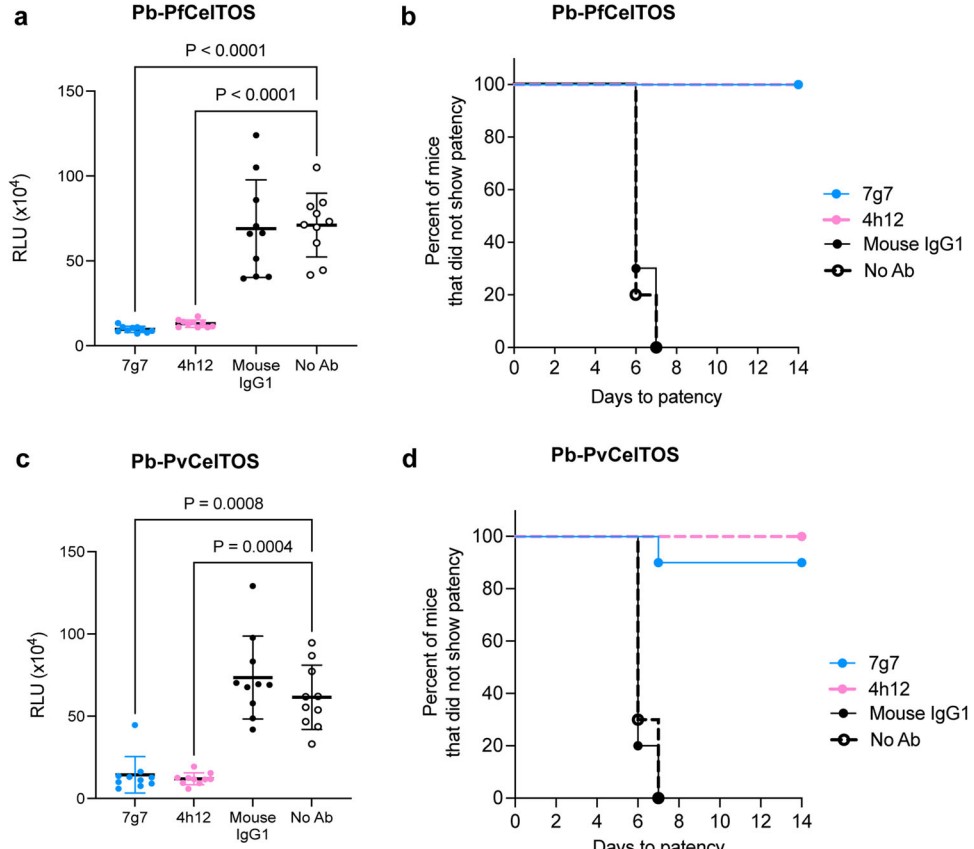

**Fig. 7 | Protective immunity of 7g7 and 4h12 in BALB/c mice.** Liver stage infectivity in mice challenged with chimeric *P. berghei* sporozoites expressing *P. falciparum* CelTOS or *P. vivax* CelTOS. **a** Pb-PfCelTOS liver burden among mice immunized with 7g7 and 4h12 mAb showed lower parasite load (RLU values) compared to naïve mice (no antibody control). The Anderson–Darling test for normality confirmed a normal distribution for Pb-PfCelTOS responses. Therefore, significance differences were determined by One-way ANOVA with a Tukey multiple comparison. Error bars represent mean ± s.d. from ten mice. **c** Similar results with lower parasite liver loads were also observed for Pb-PvCelTOS liver burden among mice immunized with 7g7 and 4h12. The Anderson–Darling test for

normality did not confirm a normal distribution for Pb-PvCelTOS responses. Therefore, significant differences were determined by Kruskal–Wallis with a Dunn's multiple comparison analysis. Antibody isotype matched control group was done using anti-PvCSP VK247 antibody 2E10.E9[38]. Error bars represent mean ± s.d. from ten mice. **b, d** The Kaplan–Meier curve showing the prepatent period of Pb-PfCelTOS and Pb-PvCelTOS sporozoites in BALB/c mice that were passively immunized with 7g7 and 4h12 mAbs. Prepatency is defined as the time to 0.1% parasitemia and statistical significance between the survival curves was assessed using the Log-Rank (Mantel–Cox) Test (*n* = 10).

## Animal study ethics statement

Mouse immunizations to generate mAbs were performed in adherence to the regulations of the Institutional Animal Care and Use Committee (IACUC) at Washington University in St. Louis under Animal Welfare Assurance #D1600245. Washington University in St. Louis has full accreditation from the Association for Assessment and Accreditation of Laboratory Animal Care International (AAALAC). All challenge studies were performed in adherence to the regulations of the Institutional Animal Care and Use Committee (IACUC) at the University of South Florida. Six- to eight-week-old female BALB/c mice (Taconic Biosciences, Rensselaer, NY) were used for antibody generation and were housed at 21 ± 2 °C, 12-h light-dark cycle with a relative humidity of 51 ± 10%. Six- to eight-week-old female Swiss Webster ND4 mice (Inotiv, West Lafayette, IN) and BALB/c mice (Inotiv, West Lafayette, IN) were used for sporozoite production and challenge experiments, respectively. These mice were housed at 21 ± 2 °C, 12-h light-dark cycle with a relative humidity of 55 ± 10%.

## Generation of mouse monoclonal antibodies against CelTOS

Monoclonal antibodies were generated at the Department of Pathology and Immunology Hybridoma Center, Washington University School of Medicine. Three BALB/c female mice (Taconic Biosciences, Rensselaer, NY) were immunized subcutaneously with 20 μg of CelTOS

proteins in emulsified complete Freund's adjuvant, followed by 20 μg of CelTOS in emulsified incomplete Freund's adjuvant, followed by one injection of CelTOS with Alhydrogel (InvivoGen, San Diego, CA). All immunizations were performed at 2-week intervals. The animal with the highest serum titer was boosted by intravenous immunization three days prior to fusion with 25 μg of CelTOS in endotoxin-free PBS. Immune splenocytes from the boosted mouse were fused to the P3X63Ag8.653 murine myeloma using standard procedures[66]. Cultures were screened initially by ELISA, and the top 20 highest CelTOS-positive polyclonal cultures were chosen for subcloning. The antigen-reactive wells were subcloned by limiting dilution twice. The final 6 and 3 monoclonal antibodies that were still positive for binding to PvCel-TOS and PfCelTOS, respectively, after subcloning were amplified, and parental stocks were frozen for storage. The final clones of each mAb were also isotyped using an ELISA kit from Thermo Fisher Scientific.

## Expression and purification of monoclonal antibodies

Monoclonal antibody cultures were added to the bottom chamber of a WHEATON® CELLine™ bioreactor flask, and CD Hybridoma media (Thermo Fisher Scientific, Waltham, MA) supplemented with non-essential amino acids, penicillin–streptomycin, sodium pyruvate, L-glutamine, and BME medium was added to the top chamber and grown at 37 °C with 5% $CO_2$. The supernatant and cells were harvested from

the bottom chamber weekly. The harvested cultures were spun down, and the cells were discarded. The IgG was then purified from the supernatant using protein A resin (Gold Biotechnology, St. Louis, MO) and IgG binding and elution buffers (Thermo Fisher Scientific, Waltham, MA). The eluted IgG was then buffer exchanged three times into PBS.

### Anti-CelTOS monoclonal antibody reactivity ELISA

All ELISAs were performed with 5 biological replicates each with 3 technical replicates. Nunc Maxisorb plates were coated overnight at 4 °C with 0.02 mg/ml PfCelTOS or PvCelTOS in carbonate buffer pH 9.6. The plates were washed three times with PBST (PBS buffer with 0.05% Tween-20). The plates were blocked with PBST with 2% BSA for one hour at room temperature. The plates were then washed three times with PBST. A tenfold dilution series of the individual mAbs in PBST with 2% BSA was added to the plate and incubated for one hour at room temperature. All mAbs, except 6c4, had a top concentration of 1 mg/ml for the dilution series. 6c4 had a top concentration of 0.1 mg/ml for this dilution series. The plates were then washed three times with PBST. A 1:5000 dilution of anti-mouse HRP-conjugated secondary antibody (Cat. # 112-035-071) (Jackson ImmunoResearch, West Grove, PA) was added to the plates and incubated for 30 min at room temperature. The plates were then washed three times with PBST. The plates were developed by adding 3,3′,5,5′-tetramethylbenzidine (TMB), and the reaction was stopped by adding sulfuric acid to a final concentration of 1 M. The final concentration of sulfuric acid in the wells was 1 M. The plates were then read using a Synergy H1 plate reader and 5gen v3.08 software, and the data were analyzed using GraphPad Prism 9 for MacOS.

### Antibody sequencing

The sequences of 4h12 and 7g7 were determined by hybridoma sequencing (GenScript, Piscataway, NJ) and de novo sequencing at the Protein Chemistry Unit, Research Technologies Branch, National Institute of Allergy and Infectious Diseases.

### Expression and purification of antibodies

Monoclonal IgG antibodies were expressed and purified from hybridoma cell lines using Protein A resin (Gold Biotechnology, St. Louis, MO).

For crystallization, the Fab fragment of 4h12 was obtained by cleaving the IgG with papain and purified with Protein A resin (Gold Biotechnology, St. Louis, MO) as previously described[67].

For the 7g7 Fab fragment, the Fab regions of the heavy and light chains of 7g7 were cloned and inserted into the vector pHLsec with a C-terminal 6-His tag on the heavy chain. The plasmids were cotransfected into Expi293 cells (Cat. #A14527) with ExpiFectamine (Thermo Fisher Scientific, Waltham, MA) according to the manufacturer's protocol and expressed as secreted proteins. The culture medium was loaded into a Ni Sepharose Excel (Cytiva, Marlborough, MA) column, and the resin was washed with Buffer A (25 mM Tris pH 7.4, 0.3 M NaCl) supplemented with 30 mM imidazole. The Fabs were eluted with Buffer A supplemented with 150 mM imidazole. The eluate was concentrated and injected onto a Superdex 75 Increase column (Cytiva, Marlborough, MA) equilibrated with 20 mM Tris. pH 8, 100 mM NaCl. Fractions were pooled and stored at −80 °C.

### Crystallization, data collection, and refinement of CelTOS-Fab complex structures

PfCelTOS was incubated with 4h12 Fab at a ratio of 1.5:1 for 30 min at room temperature to allow the formation of the PfCelTOS-4h12 complex. The complex was then purified by size exclusion chromatography using a Superdex 200 10/300 GL column (Cytiva, Marlborough, MA). Crystals of PfCelTOS-4h12 were grown at 18 °C by hanging-drop vapor diffusion after mixing 1 μl of protein at 13 mg/ml with 1 μl of reservoir

containing 0.2 M lithium acetate and 21% PEG 3350. Ethylene glycol (25%) was used as a cryoprotectant for crystal cryo-cooling. X-ray diffraction data were collected at beamline 4.2.2 of the Advanced Light Source (ALS), Berkeley National Laboratory.

PvCelTOS was incubated with 7g7 Fab at a 1:1 ratio for 30 min at room temperature to allow complex formation. The complex was then purified by size exclusion chromatography using a Superdex 200 increase 10/300 GL column (Cytiva, Marlborough, MA). Crystals of PvCelTOS-7g7Fab were grown at 17 °C by hanging-drop vapor diffusion after mixing 1 μl of protein at 15 mg/ml with 1 μl of reservoir containing 0.1 M sodium acetate, pH 4.5, and 15% PEG 4000. PEG 400 (30%) was used as a cryoprotectant for crystal cryo-cooling. X-ray diffraction data were collected at the beamline at SER-CAT 22ID beamlines, Advanced Photon Source (APS).

All diffraction data were processed with XDS[68]. The structures were solved by molecular replacement with PHASER[69] using a homology model of the variable region created by the SAbPred server[70] and the constant regions from PDB:1I3G[71] and PDB:1EMT[72] for 4h12 and PDB:6YE3[73] and PDB:5D96[74] for 7g7. Iterative model building in COOT[75] and refinement in PHENIX[76]. Data collection and refinement statistics are shown in Supplementary Table 1. Binding epitopes were identified by determining the interface between the antibody and antigen using PDBePISA[77].

### Mutation library epitope mapping using ELISA

Alanine-stretch mutations (3-4 residues) along the surface regions of CelTOS were introduced to the protein. These mutants were expressed and purified as described for the wild-type protein. ELISA was performed to determine the interaction with antibodies. Briefly, BSA or 0.02 mg/ml CelTOS variants were coated on plates overnight at 4 °C, followed by 3 washes with PBS/Tween-20 and then blocking with 2% BSA in PBS/Tween-20 for 1 h at room temperature. The plate was washed and incubated with 250 ng/ml mAb for 1 h at room temperature. The plate was then washed again and incubated with anti-mouse HRP-conjugated antibody for 30 min at room temperature. After a final wash, TMB (3,3′5,5′-tetramethylbenzidine) substrate (Sigma–Aldrich, Burlington, MA) was added for color development, and sulfuric acid was subsequently added to stop the reaction. The yellow color was measured at 450 nm.

### Determination of the binding affinity of mAb to CelTOS using BLI

All protein samples were buffer exchanged into 10 mM HEPES, pH 7.4, 150 mM NaCl, and 3 mM EDTA. The OctetRed96e system (Sartorius, Göttingen, Germany) was used to determine the kinetics of the CelTOS/mAb interactions. Briefly, mAb was immobilized onto anti-mouse IgG Fc capture (AMC) biosensors (Sartorius, Göttingen, Germany) at 25 °C. The sensors were then dipped into various concentrations of CelTOS serially diluted in HBS-EP+ buffer (Cytiva, Marlborough, MA) to measure the association, followed by dipping in HBS-EP+ buffer to measure the dissociation. Kinetic analysis of the data was conducted using Octet Data Analysis HT 12 software (Sartorius, Göttingen, Germany), and all concentrations of antigen were used for the kinetic analysis. Data were fitted globally with a 1:1 interaction model. Three biological replicates with three technical replicates each were performed for each mAb/CelTOS combination.

### Liposome disruption assay

Liposomes were prepared by dissolving lipid POPC (1-palmitoyl-2-oleoyl-sn-glycero-3-phosphocholine) and POPA (1-palmitoyl-2-oleoyl-sn-glycero-3-phosphate) (Avanti Polar Lipids, Alabaster, AL) in chloroform and then dried under nitrogen gas for 20 min. To hydrate the lipids, equal volumes of diethyl ether and 20 mM 6-carboxyfluorescein (Sigma–Aldrich, Burlington, MA) in 10 mM HEPES (pH 7.4) and 100 mM KCl were added to the lipid. Liposomes

filled with 6-carboxyfluorescein were formed through alternate cycles of vigorous shaking and an ultrasonic bath, followed by size extrusion through a Whatman® Nuclepore™ Track-Etched Membrane (pore size=200 nm) (Sigm-Aldrich, Burlington, MA). Finally, liposomes were separated from unincorporated 6-carboxyfluorescein by size exclusion chromatography using Sephadex G-25 coarse (Sigma–Aldrich, Burlington, MA)[21,22,24].

Various concentrations of Fab fragment (100, 50, 25, 12.5 or 0 nM) from different antibodies were incubated with 25 nM CelTOS for 2 hr at room temperature. Diluted liposomes were then added to a 96-well NUNC plate (Thermo Fisher Scientific, Waltham, MA), and the initial fluorescence reading ($F_{512 \text{ of liposomes}}$) was recorded (excitation = 492 nm, emission = 512 nm). Next, the protein solution was added to liposomes and incubated for 5 min at room temperature. The 6-carboxyfluorescein release from liposomes ($F_{512 \text{ of liposome+protein}}$) was then recorded. Finally, Triton-X-100 was added to complete the dequenching of liposomes ($F_{512 \text{ of liposome+triton}}$), and fluorescence was again recorded. The percent pore-forming activity was calculated as follows:

$$\% \text{ of max liposome disruption} = 100 \times \frac{F_{512 \text{ of liposome + protein}} - F_{512 \text{ of liposome}}}{F_{512 \text{ of liposome + triton}} - F_{512 \text{ of liposome}}}$$

where

$F_{512 \text{ of liposome}}$ = fluorescence reading of liposomes alone
$F_{512 \text{ of liposome+protein}}$ = fluorescence reading of liposomes + protein
$F_{512 \text{ of liposome+triton}}$ = fluorescence reading of liposomes and protein after adding Triton solution

All analyses were performed using Prism 9. The Friedman test (Dunn's multiplicity test) of nonparametric one-way ANOVA was performed. The mean values of five biological replicates and the SEM values were calculated and plotted.

### Parasite culture

*P. falciparum* KF7G4, a derivative of PfKF7 (cloned line originated from NF54, obtained from Dr. Kappe's laboratory, Seattle Biomedical Institute, USA), constitutively expresses the hdhfr-mCherry and luciferin reporter genes under the bidirectional *Pbeef1α* promoter. Using established methods[78,79], the expression cassette was integrated as part of a *piggyBac* element into a dispensable gene (PF3D7_1326900).

*P. berghei* ANKA mutant lines that were generated by replacing endogenous CelTOS with either *P. falciparum* CelTOS (Pb-PfCelTOS; 2258cl2, mutant RMgm-4066; http://www.pberghei.eu)[29] or *P. vivax* CelTOS (Pb-PvCelTOS; 2321cl3, mutant RMgm-4111; http://www.pberghei.eu)[31] were used in the challenge studies. Both transgenic *P. berghei* lines harbor a *gfp-luciferase* fusion gene under the control of the constitutive *Pbeef1α* promoter integrated into the *230p* gene locus.

### Standard membrane feeding assay (SMFA)

The ability of anti-CelTOS monoclonal antibodies to block transmission of *P. falciparum* sexual blood stages to mosquito stage oocysts was investigated with SMFA[80–82]. Briefly, a blood meal was prepared by combining *P. falciparum* PfKF7G4 gametocyte culture with fresh human erythrocytes and serum to a final gametocyte concentration of 0.3% and 50% of red blood cells in human serum. Monoclonal antibodies were added to the blood meals at a final concentration of 400 μg/ml or the same volume of phosphate buffer saline for the control and incubated at 37 °C for 30 min. The prepared blood meal at 37 °C was pipetted into a warmed hemotek blood meal reservoir and fed to female *Anopheles stephensi* mosquitoes for 30 min at room temperature. Mosquitoes were then kept in an environmental chamber at 26 °C and 80% relative humidity supplied with sugar and water ad libitum. Mosquitoes were collected 8 days after feeding for the mosquito luciferase assay. Replicates of 24, 30, 30 and 24 for groups 7g7, 4h12, no antibody and anti-DBP 3A4[37] were performed, respectively.

### Mosquito luciferase assay

Five mosquitoes were pooled as one sample in 100 μl of Passive Lysis Buffer (Promega, Madison, MI) on ice. Samples were frozen at −80 °C for 2 h and then homogenized at room temperature. The homogenates were spun at 12,000 x *g* for 2 min, and 50 μl of supernatant was transferred into a 96-well solid white plate. Steady-Glo Luciferase Assay Reagent (100 μl) was added to each sample. Luminescence was measured using a 96-well plate reader SpectraMax L Luminescence Microplate Reader (Molecular Devices, San Jose, CA) following the program of Integration time 5", PMT Autorange, Target wave 570 nm. The relative luminescence units (RLU) were recorded.

### Sporozoite production of transgenic P. berghei

Female *Anopheles stephensi* mosquitoes (3–5 days old) were fed on female Swiss Webster ND4 mice infected with either Pb-PfCelTOS[29] or Pb-PvCelTOS[31] at 2–5% parasitemia with comparable gametocytemia. The mosquitoes were maintained at 21.5 °C and 80% humidity and supplied with 10% glucose and 0.05% p-aminobenzoic acid (PABA) ad libitum. Salivary glands of infected mosquitoes were manually dissected on days 18–21 post infection and collected in L15 medium. The glands were centrifuged at 3500 x *g* for 1 min and mechanically disrupted to release the sporozoites. The sporozoite suspension was filtered using a 40 μM cell strainer and counted using a hemocytometer.

### Sporozoite Immunofluorescence assay

For indirect immunofluorescence assay, the transgenic Pb-PfCelTOS and Pb-PvCelTOS sporozoites isolated from salivary glands of infected mosquitoes were spotted at a density of $2 \times 10^4$ sporozoites/spot on a well pattern slide (Tekdon, Myakka City, FL). Cryopreserved *P. falciparum* and *P. vivax* sporozoites were thawed at 37 °C, quantified using a hemocytometer and spotted at a density of $2 \times 10^4$ sporozoites/spot on a well pattern slide. The slides were air-dried, sporozoites were fixed with 4% paraformaldehyde, permeabilized with 0.3% Triton, and blocked with 10% fetal bovine serum (FBS). The sporozoites were incubated overnight at 4 °C with 1:1000 dilution of either 7g7 or 4h12 monoclonal antibodies or 10% FBS in no primary antibody controls. The slides were washed three times with PBS followed by incubation with Alexa Fluor Plus 488 conjugated donkey anti-mouse secondary antibody (Cat. #A-2102) (Thermo Fisher Scientific, Waltham, MA) at 1:500 dilution. Nuclei were stained with Hoechst-33342 (Thermo Fisher Scientific, Waltham, MA) and coverslips were mounted using ProLong™ Glass Antifade Mountant (Thermo Fisher Scientific, Waltham, MA). Images were acquired and processed using a Zeiss LSM 900 microscope and Zen blue software v3.6.

### Sporozoite challenge study

For in vivo sporozoite challenge studies, BALB/c mice (10 mice/group) were injected intraperitonially with 580 μg of mAb 7g7, 4h12 or 2E10. E9[38] (α-*Pv*CSP VK-247, isotype control). Ninety minutes following the injection, mice were challenged with GFP-luciferase expressing 200 sporozoites of either Pb-PfCelTOS or Pb-PvCelTOS by i.v. injection. The parasite liver loads were measured at 44 hr following infection by injecting 100 μL D-luciferin (30 mg/mL), and the luciferase activity of parasites was measured using the IVIS Lumina II Imaging System (Perkin Elmer Life Sciences, Waltham, MA). Parasitemia in mice was monitored from days 4-14 post challenge by Giemsa-stained thin tail blood smears. The mice were defined as prepatent when the parasitemia reached 0.1% among RBCs counted in at least 25–30 fields at 100X magnification. The mice that did not show blood stage breakthrough until day 14 were defined as protected.

### Reporting summary

Further information on research design is available in the Nature Portfolio Reporting Summary linked to this article.

## Data availability

All data generated or analyzed during this study are included in this published article, source data file and supplementary information files. Atomic coordinates and structure factors have been deposited in the Protein Data Bank with PDB accession number 8ULF for PvCelTOS-7g7 and 8UKH for PfCelTOS-4h12 structures. Source data are provided with this paper.

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

## Acknowledgements

This work was supported by the Division of Intramural Research of the National Institute of Allergy and Infectious Diseases, National Institutes

of Health, the Extramural Research Program of the National Institutes of Allergy and Infectious Diseases, National Institutes of Health (R56 AI080792 to N.H.T and R01 AI137162 and U01 AI155361 to J.H.A.), the Burroughs Wellcome Fund (to NHT). The funders did not have any role in the study design, data collection and interpretation, or decision to submit the work for publication. We thank the Hybridoma Center and Protein Production and Purification Facility of the Rheumatic Diseases Core Center (P30AR048335) at the Washington University School of Medicine for experimental support. We thank C. Janse for generously providing the Pb-PfCelTOS or Pb-PvCelTOS transgenic parasites. We thank J. Nix and ALS Beamline 4.2.2 supported by contract DE-AC02-05CH11231 for help with X-ray diffraction data collection. We thank the staff members of the SER-CAT beamline at the Advanced Photon Source, Argonne National Laboratory for beamline support. Data were collected at the Southeast Regional Collaborative Access Team (SER-CAT) 22-ID (or 22-BM) beamline at the Advanced Photon Source, Argonne National Laboratory. SER-CAT is supported by its member institutions (see www.ser-cat.org/members.html) and equipment grants (S10_RR25528, S10_RR028976 and S10_OD027000) from the National Institutes of Health. We thank Dr. Hitoshi Otsuki of the Department of Microbiology and Immunology, Faculty of Medicine, Tottori University, Japan, for developing the luciferase reporter cassette used to develop PfKF7G4. This study used the Office of Cyber Infrastructure and Computation Biology (OCICB) High Performance Computing (HPC) cluster at the National Institute of Allergy and Infectious Diseases (NIAID), Bethesda, MD. This study used the Office of Cyber Infrastructure and Computational Biology (OCICB) High Performance Computing (HPC) cluster at the National Institute of Allergy and Infectious Diseases (NIAID), Bethesda, MD. We thank Glenn A. Nardone and Motoshi Suzuki of the Research Technologies Branch (RTB), NIAID, for de novo sequencing analysis. The following reagent was obtained from BEI Resources, NIAID, NIH: Hybridoma 2E10. E9 Anti-*Plasmodium vivax* Circumsporozoite Protein (CSP), MRA-185, contributed by Elizabeth Nardin.

## Author contributions

Conceptualization, N.H.T.; Methodology, W.T., N.D.S, S.K.K., S.X., D.V.U., H.K., J.R.J., and P.A.S.; Validation, W.T., N.D.S, S.K.K., S.X., D.V.U., H.K., and J.R.J.; Formal Analysis, W.T., N.D.S, S.K.K., S.X., D.V.U., H.K., J.R.J., J.H.A., and N.H.T.; Investigation W.T., N.D.S, S.K.K., S.X., D.V.U., H.K., J.R.J., P.A.S., and M.M.O.; Resources J.H.A. and N.H.T.; Data Curation, W.T., N.D.S, S.K.K., S.X., D.V.U., H.K., J.R.J., J.H.A., and N.H.T.; Writing – Original Draft, W.T., N.D.S., J.R.J., and N.H.T.; Writing – Review & Editing, W.T., N.D.S, S.K.K., S.X., D.V.U., H.K., J.R.J., J.H.A., and N.H.T.; Visualization, W.T., N.D.S, S.K.K., S.X., D.V.U., H.K., J.R.J., J.H.A., and N.H.T.; Supervision, J.H.A. and N.H.T.; Project Administration, S.J.B., J.H.A., and N.H.T.; Funding Acquisition, J.H.A. and N.H.T.

## Funding

## Competing interests

N.D.S, D.V.U., J.R.J, H.K., A.R., S.X., J.H.A. and N.H.T. are listed inventors on US patent US-20190276506-A1 related to this work. The remaining authors declare no competing interests.
