## [Peer Review File · Nature Communications]

Multistage protective anti-CelTOS monoclonal antibodies with cross-species sterile protection against malariaReviewers' Comments:

Reviewer #1:

Remarks to the Author:

In this work, the authors isolate a small panel of five mAbs from an immunised mouse against PvCelTOS and two against PfCelTOS. They then proceed to characterise these in detail. They determine crystal structures of their complexes with CelTOS, measure their affinities, test their effectiveness in a liposome disruption assay and assess their efficacy in a mouse model.

The structural studies are well done and reveal an interesting difference in the conformation of CelTOS in one of the complexes, which the authors suggest represents a new conformation which is relevant to pore formation. Much of CelTOS is not visualised in this structure, suggesting flexibility, but the authors have conducted mutagenesis to confirm the epitope and this structure appears well verified. These findings are interesting. The authors are right not to over speculate on what this means for CelTOS function, as this is very mysterious. However, the antibody epitope mapping seems robust.

The authors next test cross-reactivity of their antibodies to Pv and PfCelTOS. One of them, 7g7, has a 10-fold affinity different (not 100-fold, as stated in line 199). The other, 4h12, doesn't appear cross-reactive, with a very low affinity (1000-fold worse) against PvCelTOS and very poor binding curves. This finding doesn't seem entirely consistent with the subsequent mouse study in which appears to show that 4h12 does reduce liver burden in a mouse model in which PbCelTOS has been replaced by PvCelTOS. This is very curious. I also found the images of the mice in figure S2 a bit curious as they don't seem to have substantial liver loads in this experiment? Does this mean that the parasite dose was too low? Is the efficacy of 4h12 because a very high dose of antibody was used? It is strange that an antibody with such low affinity is effective in this assay. I think that the authors need to double check this, comment on why there don't seem to be parasites in the mouse not treated with antibody and comment on the mAb dose used in comparison to effective antibodies like CIS43 and L9. To me, this dataset doesn't seem internally consistent.

Finally, the authors show that the antibodies function in SMFA and prevent membrane disruption.

In summary, this is an interesting study. Only a small panel of mouse mAbs is studied, but there is plenty of useful information about the epitopes for neutralising antibodies and their ability to block transmission. The structures reveal new and intriguing insight into the CelTOS structure, while not really giving enough for a model of how CelTOS might work. The only part that I am concerned about is the cross-reactivity study, in which a very weakly binding antibody appears to function in a mouse assay.

Minor:

Lines 317-8. The set of references chosen here are rather selective, focused on the authors work. Either broaden or remove?

Line 325-327: how will these findings inform vaccine design? It seems as though the dispersed epitopes encourage use of full-length molecule. Any more that this?

Please write out methods, rather than stating 'as previously described. i.e. line 332-333.

Reviewer #2:

Remarks to the Author:

Multistage protective anti-CelTOS monoclonal antibodies with cross-species sterile protection against malaria

Summary

The manuscript titled "Multistage protective anti-CelTOS monoclonal antibodies with cross-species sterile protection against malaria" by Tang et al. describes the identification of two protective and conserved epitopes in Cell-traversal protein for ookinetes and sporozoites (CelTOS) by using two mouse monoclonal antibodies: 7g7 and 4h12. Physicochemical characterisation revealed that these mAbs inhibit CelTOS pore formation. The authors further demonstrated that these mAbs could inhibit the liver stage infection and oocyst development in mosquitoes, generating new insights into structure-based vaccine design against malaria. This is a most interesting and well conducted study we think it worthy of publication and have a few comments which will hopefully improve the manuscript.

Major Comments

1. Early preclinical studies based on CelTOS have given either partial or no protection, with direct immunisation or passive transfer of polyclonal antibodies (lines 73-82). Can the authors discuss the reasons why their approach seems to have resulted in cross species protection where the approaches have failed.
2. Epitope binning data suggested that 7g7 mAb and 8d2 mAb share an epitope, and 4h12 mAb and 4d1 mAb also share an epitopes. Monoclonal reactivity ELISA showed these mAbs have similar binding profiles. What is the authors reasoning for picking 7g7 and 4h12 rather than the other 2 for further characterisation. Was there any other biochemical information for this choice.
3. The structural data of PvCelTos-7G7 and PfCelTos-4h12 complexes are very elegant. This figure would be more informative if the detailed side chain interactions of the mAbs and the CelTos protein were added.
4. Starting at line 196 it is stated that 4h12 binds to PfCelTos with about 50 μ M which was ~1000-fold lower affinity than binding to PvCelTos. Can the authors comment on why this mAb was as efficient at inhibition of liver stage infection for CelTos from both species (Fig7). Is this why 4h12 was administered at close to half a gram per mouse? I
5. Given that 7g7 and 4h12 cross react with CelTos from both Pf and Pv, and that these mAbs are inhibitory for both in liver assays why did the authors not use 7g7 and 4h12 the cross-species liposome disruption assays? It appears that they have looked at 7g7 only in PvCelTos liposomes and 4h12 in only PfCelTos liposomes (Fig 5).
6. As this study is characterising novel conserved and conformational epitopes, it would be useful to have western blotting or immunofluorescence data on parasite material to support their

conclusions.

7. It seems there are a few inconsistencies between Fig 4a and the equivalent IVIS images in supplementary Fig2.

a. Supplementary Figure 2a, 4h12 treated group, there is an animal with significant parasite liver load (6th animal from the labelling). However, the main Figure 4a data does not appear to represent this animal correctly. In order to better understand this, it may be useful to provide the luciferase RU numbers in each animal.

8. Another potential confusing issue related to the quantitation of the luciferase assay is that 7g7 treated mice showed no evidence of luciferase activity in livers (Supp fig 2a) whereas 4h12 treated mice did. However in the blood stage assay 100% of mice treated with 4h12 showed 100% sterile protection and 90% of mice treated with 7g7 showed sterile protection. Could the authors comment or clarify this.

9. As you have measured the parasitemia in each group until Day 14, we think it would be better if you could add graphs with parasitemia fluctuations in each group to understand the level of liver breakthrough in each group and the impact of mAbs inhibiting the liver stage of the parasite.

10.

Minor Comments

1. Remove “the” between These and two on line 116

2. Monoclonal antibody reactivity ELISA, units in X axis may not be right. Should the x axis be mg/ml instead μM ? In the methods, the authors stated that the mAb concentrations started with 1mg/ml ($\sim 6.66 \mu\text{M}$). But data points are located at 100, 10⁻¹, 10⁻², 10⁻³, 10⁻⁴, 10⁻⁵ and 10⁻⁶ μM .

3. Please describe what is represented by error bars in Figure 2f legend.

4. Please correct the lettering of Figure 3e panel it appears that there are two panels labelled D

5. In the “4h12 traps CelTOS in a unique oligomeric state with reordered dimer interface” line 175 subsection of the results. It seems that Fig.1c should in fact be Fig 2c

Reviewer #3:

Remarks to the Author:

RESPONSE TO REVIEWER COMMENTS

Reviewer #1 (Remarks to the Author):

In this work, the authors isolate a small panel of five mAbs from an immunised mouse against PvCelTOS and two against PfCelTOS. They then proceed to characterise these in detail. They determine crystal structures of their complexes with CelTOS, measure their affinities, test their effectiveness in a liposome disruption assay and assess their efficacy in a mouse model.

The structural studies are well done and reveal an interesting difference in the conformation of CelTOS in one of the complexes, which the authors suggest represents a new conformation which is relevant to pore formation. Much of CelTOS is not visualised in this structure, suggesting flexibility, but the authors have conducted mutagenesis to confirm the epitope and this structure appears well verified. These findings are interesting. The authors are right not to over speculate on what this means for CelTOS function, as this is very mysterious. However, the antibody epitope mapping seems robust.

Response: We thank the reviewer for their positive comments.

The authors next test cross-reactivity of their antibodies to Pv and PfCelTOS. One of them, 7g7, has a 10-fold affinity different (not 100-fold, as stated in line 199).

Response: We thank the reviewer for identifying an error in the original manuscript. This has corrected to “10-fold” in the revised manuscript.

The other, 4h12, doesn't appear cross-reactive, with a very low affinity (1000-fold worse) against PvCelTOS and very poor binding curves. This finding doesn't seem entirely consistent with the subsequent mouse study in which appears to show that 4h12 does reduce liver burden in a mouse model in which PbCelTOS has been replaced by PvCelTOS. This is very curious. I also found the images of the mice in figure S2 a bit curious as they don't seem to have substantial liver loads in this experiment? Does this mean that the parasite dose was too low? Is the efficacy of 4h12 because a very high dose of antibody was used? It is strange that an antibody with such low affinity is effective in this assay. I think that the authors need to double check this, comment on why there don't seem to be parasites in the mouse not treated with antibody and comment on the mAb dose used in comparison to effective antibodies like CIS43 and L9. To me, this dataset doesn't seem internally consistent.

Response: We thank the reviewer for identifying this issue. We do agree with the reviewers that the parasite liver loads in the control groups were less than expected. We investigated this issue and observed the inconsistent parasite liver burdens with a few other unrelated experiments performed around the same time as these challenges, and

subsequently identified that the stock of D-luciferin used for *in vivo* imaging during this time period was defective and caused these discrepancies. We therefore repeated the entire challenge study with fresh lot of D-luciferin substrate and obtained publication quality data. The protective outcomes were comparable to the prior results presented in the original manuscript with much more robust *in vivo* imaging data. We replaced the original data with the new replicated results in the revised manuscript. It is important to note that this is an entire second biological replicate to validate the reproducibility that a single antibody dose of either 7g7 or 4h12 protects against Pf and Pv transgenic parasites. In Supplementary Fig.3, the new data reveals that the control groups showed substantial liver loads and the luciferase signals were significantly higher in the control groups than in the presence of either 7g7 or 4h12 (Fig. 7). Readings of the luciferase signal of each mouse were reported in Supplementary Fig.3.

We apologize to this reviewer that the amount of antibody used for the challenge was erroneously reported. Instead of “580 μ g”, we reported “580 mg”. This correction has been made in the revised version and we apologize in giving the wrong impression that a high dose of antibody was used.

Finally, the authors show that the antibodies function in SMFA and prevent membrane disruption.

In summary, this is an interesting study. Only a small panel of mouse mAbs is studied, but there is plenty of useful information about the epitopes for neutralising antibodies and their ability to block transmission. The structures reveal new and intriguing insight into the CelTOS structure, while not really giving enough for a model of how CelTOS might work.

Response: We thank the reviewer for their positive comments.

The only part that I am concerned about is the cross-reactivity study, in which a very weakly binding antibody appears to function in a mouse assay.

Response: We agree with the reviewer that there is a discrepancy between the affinity measurements and *in vivo* protection data. We have now repeated the challenges studies multiple independent times and reproducibly obtained complete sterile protection against both Pf and Pv transgenic parasites. The protection data is robust. We have also demonstrated in this manuscript that CelTOS is conformational flexible. We believe the architecture of CelTOS are very different between the pre-pore state and pore state. The binding affinities we reported here can only reflect the interaction with the CelTOS dimer. It is believed that CelTOS is arranged in an array on the surface of the parasite (Kariu et. al., 2006; Bergmann-Leitner et. al., 2010), and this array would enhance the binding affinity of the antibody to the parasite by avidity. This enhancement would not be captured by a binding study to the dimer, and the array cannot be adequately captured in a form for study using accurate quantitation of binding parameters.

In the revised manuscript we have also added IFA studies to investigate cross-species binding of the antibodies directly to parasites and the data is reported in Fig. 5c-f. The results demonstrated that both 7g7 and 4h12 recognized CelTOS proteins in transgenic *P. berghei* parasites expressing either Pf or Pv CelTOS, and to natural *P. falciparum* and *P. vivax* sporozoites. These data validate the cross-species binding of the antibodies directly to parasites and the binding is likely driven by the array presentation of CelTOS over the entire surface of the parasite.

Minor:

1. *Lines 317-8. The set of references chosen here are rather selective, focused on the authors work. Either broaden or remove?*

Response: We have edited the citations so they are all relevant to the content and from multiple groups.

2. *Line 325-327: how will these findings inform vaccine design? It seems as though the dispersed epitopes encourage use of full-length molecule. Any more that this?*

Response: The discussion includes the statement "The disparate locations of the 7g7 and 4h12 epitopes in CelTOS suggested that the full length CelTOS protein should be prioritized for further development and that focusing the immune response to one epitope would eliminate the benefits of retaining the second epitope". CelTOS is a ~18kDa protein and this is the first report to show the binding epitopes at atomic resolutions of neutralization antibodies. With these findings, future studies using full-length CelTOS, truncations of CelTOS, CelTOS peptides or other protein engineering approaches are now possible to improve CelTOS as a vaccine candidate. However, these studies are beyond the scope of this manuscript and speculating on the best approach in this study is not suitable without extensive evaluation.

3. *Please write out methods, rather than stating 'as previously described. i.e. line 332-333.*

Response: The method for expression and purification of CelTOS has written out in details in Methods section.

Reviewer #2 (Remarks to the Author):

Multistage protective anti-CeLTOS monoclonal antibodies with cross-species sterile protection against malaria

Summary

The manuscript titled “Multistage protective anti-CeLTOS monoclonal antibodies with cross-species sterile protection against malaria” by Tang et al. describes the identification of two protective and conserved epitopes in Cell-traversal protein for ookinetes and sporozoites (CeLTOS) by using two mouse monoclonal antibodies: 7g7 and 4h12. Physicochemical characterisation revealed that these mAbs inhibit CeLTOS pore formation. The authors further demonstrated that these mAbs could inhibit the liver stage infection and oocyst development in mosquitoes, generating new insights into structure-based vaccine design against malaria. This is a most interesting and well conducted study we think it worthy of publication and have a few comments which will hopefully improve the manuscript.

Response: We thank the reviewer for their positive comments.

Major Comments

1. Early preclinical studies based on CeLTOS have given either partial or no protection, with direct immunisation or passive transfer of polyclonal antibodies (lines 73-82). Can the authors discuss the reasons why their approach seems to have resulted in cross species protection where the approaches have failed.

Response: Immunizing animals with CeLTOS as vaccine will trigger the immune system to elicit polyclonal antibodies against the entirety of the immunogen. In both vaccination and passive transfer of polyclonal Ab, a mixture of both functional and non-functional antibodies is present that recognized CeLTOS. The failure in cross-species protection or effective protection by those approaches could be due to the low proportion of functional antibodies that present over other non-functional antibodies. Secondly, this study is the first to clearly demonstrate that CeLTOS undergoes multiple conformations in solution. Immunization with an immunogen that exhibits multiple conformations may result in a poor immune response as all conformations are being recognized therefore preventing potent recognition of the protective state. In our current study, we passively transferred only a single functional mAb and therefore the protection is clear.

2. Epitope binning data suggested that 7g7 mAb and 8d2 mAb share an epitope, and 4h12 mAb and 4d1 mAb also share an epitopes. Monoclonal reactivity ELISA showed these mAbs have similar binding profiles. What is the authors reasoning for picking 7g7 and 4h12 rather than the other 2 for further characterisation. Was there any other biochemical information for this choice.

Response: Our initial selection process was mainly based on the stability and expression yield of individual hybridoma cells, followed by whether the antibodies could

be crystallized with CelTOS to provide atomic details on the binding epitope. Based on these criteria, 4d1 was not selected as the antibody was not stably expressed. We tried to crystallize both 7g7 and 8d2, and we were able to obtain crystals for 7g7 but not 8d2. As 7g7 and 8d2 shares overlapping epitope, we decided to focus on 7g7.

3. The structural data of PvCelTos-7G7 and PfCelTos-4h12 complexes are very elegant. This figure would be more informative if the detailed side chain interactions of the mAbs and the CelTos protein were added.

Response: We thank the reviewer for this suggestion to improve the manuscript. We have added Supplementary Fig. 1 to illustrate the interactions between CelTOS and antibodies, in addition to the tablet format in Supplementary Table 2 & 3.

4. Starting at line 196 it is stated that 4h12 binds to PfCelTos with about 50 μ M which was ~1000-fold lower affinity than binding to PvCelTos. Can the authors comment on why this mAb was as efficient at inhibition of liver stage infection for CelTos from both species (Fig7). Is this why 4h12 was administered at close to half a gram per mouse? I

Response: We apologize to this reviewer that the amount of antibody used for the challenge was erroneously reported. Instead of “580 μ g”, we reported “580 mg”. This correction has been made in the revised version and we apologize in giving the wrong impression that a high dose of antibody was used.

5. Given that 7g7 and 4h12 cross react with CelTos from both Pf and Pv, and that these mAbs are inhibitory for both in liver assays why did the authors not use 7g7 and 4h12 the cross-species liposome disruption assays? It appears that they have looked at 7g7 only in PvCelTos liposomes and 4h12 in only PfCelTos liposomes (Fig 5).

Response: CelTOS-mediated membrane disruption assay was used as an initial screening assay to evaluate the in vitro functional activity of isolated antibodies. At that early stage, we did not think of cross-species capabilities and therefore do not have the data.

6. As this study is characterising novel conserved and conformational epitopes, it would be useful to have western blotting or immunofluorescence data on parasite material to support their conclusions.

Response: We thank the reviewer for identifying methods to improve the manuscript. We performed the requested sporozoite IFA using both 4h12 and 7g7 mAbs on both Pb-PfCelTOS and Pb-PvCelTOS transgenic sporozoites as well as cryopreserved *P. falciparum* and *P. vivax* sporozoites and presented the data in Figure 5 of the revised manuscript. The results demonstrated that both 7g7 and 4h12 recognized CelTOS proteins in transgenic *P. berghei* parasites expressing either Pf or Pv CelTOS, and to natural *P. falciparum* and *P. vivax* sporozoites. These data validate the cross-species

binding of the antibodies directly to parasites and the binding is likely driven by the array presentation of CelTOS over the entire surface of the parasite. CelTOS is conformational flexible and can induce pore formation. We believe the architecture of CelTOS are very different between the pre-pore state and pore state. The binding affinities we reported here can only reflect the interaction with CelTOS dimer. It is believed that CelTOS is arranged in an array on the surface of the parasite (Kariu et. al., 2006; Bergmann-Leitner et. al., 2010) and this array would enhance the binding affinity of the antibody to the parasite by avidity. This enhancement would not be captured by a binding study to the dimer, and the array cannot be adequately captured in a form for study using accurate quantitation of binding parameters.

7. It seems there are a few inconsistencies between Fig 4a and the equivalent IVIS images in supplementary Fig2a. Supplementary Figure 2a, 4h12 treated group, there is an animal with significant parasite liver load (6th animal from the labelling). However, the main Figure 4a data does not appear to represent this animal correctly. In order to better understand this, it may be useful to provide the luciferase RU numbers in each animal.

Response: We thank the reviewers for this comment. We apologize for the confusion caused by the mice *in vivo* images and RLUs in the original manuscript. As requested by multiple reviewers, we repeated the entire challenge study, and the new data was reported in this revised version. The new data corrects the issue raised by this comment. The protective outcomes were comparable to the prior results presented in the original manuscript with much more robust *in vivo* imaging data. We replaced the original data with the new replicated results in the revised manuscript. It is important to note that this is an entire second biological replicate to validate the reproducibility that a single antibody dose of either 7g7 or 4h12 protects against Pf and Pv transgenic parasites. In Supplementary Fig.3, the new data reveals that the control groups showed substantial liver loads and the luciferase signals were significantly higher in the control groups than in the presence of either 7g7 or 4h12 (Fig. 7). As recommended by the reviewer, readings of the luciferase signal of each mouse were reported in Supplementary Fig.3.

8. Another potential confusing issue related to the quantitation of the luciferase assay is that 7g7 treated mice showed no evidence of luciferase activity in livers (Supp fig 2a) whereas 4h12 treated mice did. However in the blood stage assay 100% of mice treated with 4h12 showed 100% sterile protection and 90% of mice treated with 7g7 showed sterile protection. Could the authors comment or clarify this.

Response: We apologize for the confusion caused by the mice *in vivo* images and RLUs. We do agree with the reviewers that the parasite liver loads in the control groups were less than expected. We investigated this issue and observed the inconsistent parasite liver burdens with a few other unrelated experiments performed around the same time as these challenges, and subsequently identified that the stock of D-luciferin used for *in vivo*

imaging during this time period was defective and caused these discrepancies. We therefore repeated the entire challenge study with fresh lot of D-luciferin substrate and obtained publication quality data. The protective outcomes were comparable to the prior results presented in the original manuscript with much more robust *in vivo* imaging data. We replaced the original data with the new replicated results in the revised manuscript. It is important to note that this is an entirely second biological replicate to validate the reproducibility that a single antibody dose of either 7g7 or 4h12 protects against Pf and Pv transgenic parasites. In Supplementary Fig.3, the new data reveals that the control groups showed substantial liver loads and the luciferase signals were significantly higher in the control groups than in the presence of either 7g7 or 4h12 (Fig. 7). Readings of the luciferase signal of each mouse were reported in Supplementary Fig.3.

9. As you have measured the parasitemia in each group until Day 14, we think it would be better if you could add graphs with parasitemia fluctuations in each group to understand the level of liver breakthrough in each group and the impact of mAbs inhibiting the liver stage of the parasite.

Response: The goal of the experiment is to evaluate sterile protection induced by the passive administration of monoclonal antibodies 7g7 and 4h12 before the sporozoite challenge. Sporozoite infection of a liver cell in the *Plasmodium* life cycle is a bottleneck as a single liver schizont can result in thousands of first-generation merozoites that are released into the blood and initiate an erythrocytic cycle (Prudencio *et. al.*, 2006; PMID: 17041632). We evaluated the blood stage breakthrough following the pre-patency period which is determined as the time between sporozoite inoculation and the first appearance of blood-stage parasites. To minimize the variability in counting the pre-patency period among the different animals, we kept a cut-off of 0.1% parasitemia that was counted at least among 15000 RBCs that were spread across 30 fields. Once the mouse showed 0.1% parasitemia, it is defined as not protected and is removed from the experiment. Tail blood smears were evaluated from the mice that were negative for blood stage breakthrough and were monitored for any blood stage parasites until day 14 post-challenge. Lastly, there is no delay in parasitemia and very few mice demonstrated breakthrough, therefore quantitation of parasitemia will not provide substantially more information that the sterile protection presented.

Minor Comments

1. Remove "the" between These and two on line 116

Response: We thank the reviewer for identifying errors in the manuscript. This error has been removed.

2. Monoclonal antibody reactivity ELISA, units in X axis may not be right. Should the x axis be mg/ml instead μ M? In the methods, the authors stated that the mAb concentrations started with 1mg/ml (\sim 6.66 μ M). But data points are located at 100, 10-1, 10-2, 10-3, 10-4, 10-5 and 10-6 μ M.

Response: We thank the reviewer for this constructive comment and have revised the x-axes in the plots from μM to mg/ml . The description in Methods is correct and we have revised the axis in the figure.

3. Please describe what is represented by error bars in Figure 2f legend.

Response: We added “The graph is shown as means \pm SDs of three biological replicates.” in the legend of 2f and 3e to describe error bars.

4. Please correct the lettering of Figure 3e panel it appears that there are two panels labelled D

Response: We thank the reviewer for identifying this error and the label has been revised accordingly.

5. In the “4h12 traps CeITOS in a unique oligomeric state with reordered dimer interface” line 175 subsection of the results. It seems that Fig.1c should in fact be Fig 2c

Response: We thank the reviewer for identifying an error that has been revised to Fig 2c.

Reviewer #3 (Remarks to the Author):

We thank this reviewer for their constructive comments above.

Reviewers' Comments:

Reviewer #1:

Remarks to the Author:

The manuscript is much improved and most of my comments have been very well addressed. In particular, the new mouse data is much better. I think that the comments about the apparent conflict between the mouse experiment and antibody affinity in the response to reviewers was entirely sensible. It would be nice to have a few sentences about this in the discussion. I would also remove statements about structure-guided vaccine design without any particular conclusions being stated in the manuscript about what these are. But happy to let that pass if the authors prefer to make these broad statements.

Reviewer #2:

Remarks to the Author:

We have read the revised manuscript and commend the authors for effectively addressing our initial comments and revising the manuscript accordingly. As a result, we find the manuscript suitable for publication.

We have some suggestions for future research in this area. The authors hypothesise that the discrepancy between in-vitro affinity measurements and in-vivo protection data is primarily due to CelTOS being arranged in an array on the parasite surface, which enhances binding affinity through the avidity effect. We agree with the authors that binding studies on the dimer might not capture this avidity enhancement using standard techniques like SPR, BLI, MST, or ITC. To test this hypothesis in the future, the authors could employ simple ELISA techniques, such as Capture ELISA for CelTOS, probing with HRP-conjugated CelTOS mAbs. While ELISA may not provide absolute affinity values, it can offer valuable insights into the hypothesis. Given the authors' hypothesis, they might also consider performing IFA studies on unpermeabilised sporozoites and using super-resolution (air scan) imaging to provide more compelling evidence to the readers. As this is apparently evident in Bergmann-Leitner et al., 2010 paper, we do not think this is necessary for this manuscript.

Reviewer #3:

Remarks to the Author:

I co-reviewed this manuscript with one of the reviewers who provided the listed reports. This is part of Nature Communications initiative to facilitate peer review training and provide appropriate recognition for Early Career Researchers who co-review manuscripts.

RESPONSE TO REVIEWER COMMENTS

Reviewer #1 (Remarks to the Author):

The manuscript is much improved and most of my comments have been very well addressed. In particular, the new mouse data is much better. I think that the comments about the apparent conflict between the mouse experiment and antibody affinity in the response to reviewers was entirely sensible. It would be nice to have a few sentences about this in the discussion. I would also remove statements about structure-guided vaccine design without any particular conclusions being stated in the manuscript about what these are. But happy to let that pass if the authors prefer to make these broad statements.

Response: We thank the reviewer for their positive comments. We elected not to include the suggested optional changes to the manuscript, particularly as we do not have concrete evidence for the cause of the apparent discrepancy in affinity experiments vs *in vivo* mouse studies.

Reviewer #2 (Remarks to the Author):

We have read the revised manuscript and commend the authors for effectively addressing our initial comments and revising the manuscript accordingly. As a result, we find the manuscript suitable for publication.

Response: We thank the reviewer for their positive comments and for recommending publication.

*We have some suggestions for future research in this area. The authors hypothesise that the discrepancy between *in-vitro* affinity measurements and *in-vivo* protection data is primarily due to CelTOS being arranged in an array on the parasite surface, which enhances binding affinity through the avidity effect. We agree with the authors that binding studies on the dimer might not capture this avidity enhancement using standard techniques like SPR, BLI, MST, or ITC. To test this hypothesis in the future, the authors could employ simple ELISA techniques, such as Capture ELISA for CelTOS, probing with HRP-conjugated CelTOS mAbs. While ELISA may not provide absolute affinity values, it can offer valuable insights into the hypothesis. Given the authors' hypothesis, they might also consider performing IFA studies on unpermeabilised sporozoites and using super-resolution (air scan) imaging to provide more compelling evidence to the readers. As this is apparently evident in Bergmann-Leitner et al., 2010 paper, we do not think this is necessary for this manuscript.*

Response: We thank the reviewer for their insightful comments on future directions. The approaches proposed and additional studies would form an interesting follow-on study that we may consider in the future.

Reviewer #3 (Remarks to the Author):

I co-reviewed this manuscript with one of the reviewers who provided the listed reports. This is part of Nature Communications initiative to facilitate peer review training and provide appropriate recognition for Early Career Researchers who co-review manuscripts.

Response: We thank this reviewer for their constructive comments above.